

# The Glaciers of the Dolomites: last 40 years of melting

Andrea Securo [1,2,3], Costanza Del Gobbo [4], Giovanni Baccolo [5,3], Carlo Barbante [1,2,3],
Michele Citterio [6,3], Fabrizio De Blasi [2,1,3], Marco Marcer [7], Mauro Valt [8], and Renato R. Colucci [2,3,4]

[1] Department of Environmental Sciences, Informatics and Statistics, University Ca Foscari of Venice
[2] Institute of Polar Sciences, National Research Council of Italy
[3] Comitato Glaciologico Italiano
[4] Alpine-Adriatic Meteorological Society
[5] Department of Science, Roma Tre University
[6] Department of Glaciology and Climate, Geological Survey of Denmark and Greenland
[7] Department of Environmental and Resource Engineering, Geotechnics & Geology, Technical University of Denmark
[8] Agency for Environmental Prevention and Protection of the Veneto Region

**Correspondence:** Andrea Securo (andrea.securo@unive.it)

**Abstract.** Small Alpine glaciers located below the regional equilibrium line altitude are experiencing considerable ice losses and are expected to fragment into smaller glacial bodies and eventually disappear. Monitoring such glaciers through remote sensing is often challenging because of the incompatibility between their size and the spatial resolution of satellites. The Italian Dolomites (S-E Alps) are a region clearly illustrating such challenges and where no long-term glacier mass balance data are

available. This renowned Alpine sector hosted tens of glaciers up until a few decades ago, with now only twelve remaining. This study presents a multi-decadal (1980s-2023) estimation of surface elevation change and geodetic mass balance of the current mountain glaciers present in the area. Calculations are based on geodetic data: high resolution and accuracy is obtained with unmanned aerial vehicle (UAV) Structure from Motion (SfM) and airborne Light Detection and Ranging (LiDAR), from 2010 to 2023. SfM on historical aerial imagery is used for previous decades. We found an average cumulative surface elevation

change of -28.7 m from 1980s to 2023, 33% of which between 2010-2023. The average mass balance rate for the whole period is -0.64 ± 0.05 m w.e. yr$^{-1}$, varies significantly between sites, and is negative with a smaller amplitude than the Alpine reference glaciers mass balance. Regionally, 66% of the entire volume loss is related to the *Marmolada* Glacier alone. Mass losses are accompanied by areal reductions evidencing that the Dolomites are rapidly losing their glaciers. This study aims to address the existing lack of multi-decadal data for the Dolomites by providing a quantitative account of the current state of these small

glacial bodies.

## 1 Introduction

Glaciers worldwide have been losing mass at alarming rates over the past decades (Zemp et al., 2019; Hugonnet et al., 2021), with the greatest emphasis on regions of the world that are warming faster than average (Rantanen et al., 2022; ICCI, 2022). Among these regions are the European Alps (Hock et al., 2019), where severe impacts on the cryosphere have been observed

because of rising temperature (Auer et al., 2007), changes in seasonal precipitation patterns, total radiation, humidity, and snow accumulation (Gobiet et al., 2014; Rumpf et al., 2022).



The European Alps can effectively be considered the birthplace of modern glaciology (Clarke, 1987). Despite the progressive improvement of *in-situ* (Huss et al., 2015; WGMS, 2021) and remotely sensed cryosphere monitoring efforts for the European Alps (Davaze et al., 2020; Paul et al., 2020; Sommer et al., 2020), areas remain where only scattered data and observations are available. One of those regions are the Italian Dolomites (S-E Alps).

The Dolomites are mostly known for their landscapes, resulting from an extremely rich geological diversity (Panizza, 2009). Their sceneries and relatively easy accessibility made them one of the most important mountain tourism hubs in the entire Alps, both in winter and summer (Bertocchi et al., 2021). At the same time, the area has been permanently inhabited for thousands of years, making human presence a dominant factor in defining the landscape, especially in the main valley bottoms (Broglio, 2016). The Dolomites represent an extremely complex area from a socio-economic, geographical and cultural point of view, and for this reason have been studied intensively in terms of geology, geomorphology, and biodiversity (Bosellini et al., 2003; Panizza, 2009; Pignatti and Pignatti, 2014). However, glaciers of this region have always remained on the margins of scientific exploration.

Glaciers in the Dolomites region were numerous until a few decades ago (CGI-CNR, 1962). The complex topography of the area ensures that many sites are orographically protected from direct sunlight and favour the accumulation of snow through avalanches. Until recently, such a context allowed the existence of glaciers well-below the environmental equilibrium line altitude (envELA) (Castiglioni, 1925; Žebre et al., 2021), which in the Eastern Alps during the 20th century ranged from 2700 to 3100 m a.s.l. (Žebre et al., 2021). Only a few massifs had an accumulation basin located above the envELA, allowing larger glaciers to develop, such as the *Marmolada* and *Fradusta* Glaciers, which until recently were the two largest ice bodies of the Dolomites. In the 1960s, the surfaces of these two glaciers alone were 3.05 and 0.65 km$^2$ respectively, representing 45% of the total former Dolomites glaciers surface (CGI-CNR, 1962).

Despite the information about the glaciers of the Dolomites available from the two releases of the registers of Italian glaciers (CGI-CNR, 1962; Smiraglia et al., 2015), data remains extremely scattered. More accurate information is available for the *Marmolada* Glacier, by far the largest in the area and the only one receiving scientific attention (Santin et al., 2019). The deadly ice avalanche that occurred on this glacier in July 2022, resulting in 11 fatalities, has created an additional heightened interest (Bondesan and Francese, 2023).

Assessing the state of glaciers through their mass balance is considered an essential component in the global climate-related set of observations (Zemp et al., 2009). At the regional scale, this involves generating large composite datasets through *in-situ* measurements and remote sensing campaigns. Various techniques have been developed over the last few decades to improve remote measurements of glacier mass change (Berthier et al., 2023). Workflows involving sensors mounted on unmanned aerial vehicles (UAVs) or helicopters are particularly effective for monitoring smaller topography-bounded glaciers. Among such settings, one technique that has been successfully used is Structure from Motion (SfM) (Carrivick et al., 2016). SfM allows both terrestrial (e.g., Piermattei et al., 2015; Marcer et al., 2017) and aerial (e.g., Smith et al., 2016) high-resolution surveys relying on photos taken from different positions. The same workflow can be applied to historical imagery initially acquired for mapping purposes, allowing for the retrieval of past glacier change (Mertes et al., 2017; Knuth et al., 2023). Despite the large improvements in the applications of SfM, the state-of-the-art for high-resolution topographic surveys remains





Light Detection and Ranging (LiDAR) (Bhardwaj et al., 2016). LiDAR is an active remote sensing technique that uses a laser pulse and its two-way travel time to reconstruct the surrounding scene. It can be used to monitor surface elevation changes at different scales, depending on the sensor and platform used (e.g., Knoll and Kerschner, 2009; Fischer et al., 2016; Okyay et al., 2019; Securo et al., 2022).

To the best of our knowledge, no specific studies have been conducted regarding the current status of Dolomites glaciers, or about their decadal change. The aim of our work is to fill this gap, providing a description of the glaciers in the Dolomites that are still active, and quantifying their evolution during the last 40 years. To reach these objectives, surface elevation changes and mass balance across the 12 current Dolomites mountain glaciers have been computed from the 1980s to 2023, with almost one measure per decade. Available temperature, precipitation and snow accumulation data from the regional network of weather stations is used to contextualize the results.

## 2   Previous glaciological research in the Dolomites

Glaciological observations in the Dolomites began in the late 19[th] century but were only sporadically conducted across the 20[th] century and in more recent decades. No glacier in the area has mid or long-term mass balance dataset available.

The first study to mention the Dolomites glaciers dates back to 1888 (Richter). In his volume, the Austrian glaciologist described the then-known glaciers in the eastern Alpine region, including two of the largest ice bodies of the Dolomites: *Marmolada* and *Travignolo* Glaciers. However, the first work introducing topographical maps and quantitative information was compiled by *Olinto Marinelli* (1910). The Italian geographer meticulously analysed 39 glaciers in the Dolomites region, visiting them between 1893 and 1909. He made descriptions of many glacial bodies previously unknown and made an accurate analysis to distinguish glaciers from bare ice deposits, providing a robust overview of the cryosphere state during the early 20[th] century. Thereafter, the available information about the glaciers of the Dolomites mostly comes from works that have dealt generally with glaciers in the Italian Alps, such as the releases of the Italian inventory of Alpine glaciers (Porro, 1925; CGI-CNR, 1962; Smiraglia et al., 2015).

Data presented in wider Italian Alps or Pan-Alpine products is not always accurate for the Dolomites. For example, some glaciers reported as extinct in the 1962 inventory (CGI-CNR), are still measurable and existing according to latest inventory (Smiraglia et al., 2015). In the past, only sporadic attention, from a glaciologic point of view, has been paid to specific massifs of the Dolomites (Castiglioni, 1925, 1930; Nangeroni, 1938; Del Longo et al., 2001; Cibien et al., 2007). Some more information is available from institutional reports edited for administrative regions, but considering the incompleteness of these dispatches, they are not discussed here. According to the national Italian inventory (CGI-CNR, 1962), in the late 1950's, the Dolomites were hosting 33 glaciers, with a total surface of 8.19 km$^2$.

The last partial survey of the Dolomites glaciers area was carried out by the Regional Environmental Protection Agency of the Veneto region (ARPAV), which analysed the surface variations for 27 main glacial bodies of the Dolomites (Crepaz et al., 2013), representing the 72% of the total glacierised area in the region. Results show an area variation of approximately -50% from 1910 to 2009. The most recent inventory available for Italian glaciers (Smiraglia et al., 2015) reports that 51 glacial bodies





90    were present in the Dolomites in 2009, spanning a total area of 5.04 km$^2$, making up 1.4% of the total Italian glacierised area. Among the 51 glacial bodies, 13 are classified as mountain glaciers (Table 1) while 38 are considered snow or ice patches. Also of great significance is the presence of debris coverage, which is abundant or complete on 18 glacial bodies. When we use the term ice patch, we refer to the description of ice patch of glacial origin present in Serrano et al. (2011), which is more specific and relevant to the study area compared to the definition of dead ice.

95    The only glacier of the Dolomites that has been extensively studied in the past years is the largest in the area, the *Marmolada Principale* Glacier. It was surveyed with ground-penetrating radar in 2004 and 2014, revealing a 30% reduction in thickness, hinting at an imminent fragmentation and a likely complete disappearance before 2050 (Santin et al., 2019). The transition between glacial and periglacial landforms in the area is discussed in Seppi et al. (2014), while studies focused on rock glaciers (Krainer et al., 2010, 2012) and debris covered glaciers (Securo et al., 2024a) have also been done recently. Despite their

100   limited size, the glaciers and glacial bodies in the Dolomites have the potential to cause geohazards. This is first-handedly confirmed by the recent Marmolada ice and rock avalanche (3 July 2022, Bondesan and Francese, 2023), but also by the debris flow events observed on *Monte Pelmo*, closely related to the *Val D'Arcia* glacial body (Del Longo et al., 2001; Chiarle et al., 2007) and by the instability of some little ice age moraines (Zanoner et al., 2017).

To the aims of the present study, we used names and sites mentioned in the Smiraglia et al. (2015) inventory. As this work

105   specifically focuses on the remaining mountain glaciers of the Dolomites area, attention is given to the following mountain ranges (Fig. 1): *Popera, Cristallo, Sorapiss, Antelao, Marmolada* and *Pale di San Martino*. Other Dolomites massifs that still host minor ice deposits devoid of any evidence of dynamics are not included in this work. *Fradusta* and *Marmolada* Glaciers, which recently experienced fragmentation, will still be treated as single glaciers. With these criteria, 9 glaciers are involved in the present study.





**Figure 1.** Dolomites in the European Alps (©*Map Tiler - © OpenStreetMap contributors 2023. Distributed under the Open Data Commons Open Database License (ODbL) v1.0.*) (a). Position of the 6 mountain areas hosting active glaciers in the Dolomites, location of snow ice patches (white dots), mountain glaciers (light-blue dots) (Smiraglia et al., 2015), Automatic Weather Stations (AWS) (yellow) and snow depth observations stations (red) (b). 3D views of the Dolomites glaciers using LiDAR data from 2014 (c). A few AWS used in this study are located outside of map b.



| (Sector ID) Name | RGI-ID [a] | CGI-ID | Elevation [a] | Aspect | Area [b] |
|---|---|---|---|---|---|
| *(1) Popera Alto* | -11.03918 | 978 | 2486 | E | 0.09 |
| *(1) Popera Pensile* | -11.03919 | 977 | 2722 | N-E | 0.07 |
| *(2) Cristallo* | -11.03903 | 937 | 2483 | N | 0.24 |
| *(3) Sorapiss Occidentale* | -11.03902 | 975 | 2661 | N | 0.19 |
| *(4) Antelao Superiore* | -11.03910 | 966 | 2617 | N-E | 0.27 |
| *(4) Antelao Inferiore* | -11.03909 | 967 | 2441 | N | 0.19 |
| *(5) Marmolada* [c] | - | - | 2918 | N | 1.49 |
| *M. Principale* | -11.03887 | 941 | - | - | - |
| *M. Punta Penia* | -11.03883 | 942 | - | - | - |
| *M. Ovest* | -11.03880 | - | - | - | - |
| *M. Centrale* | -11.03884 | - | - | - | - |
| *(6) Fradusta* [d] | - | 950 | 2703 | N | 0.03 |
| *F. Superiore and Inferiore* | -11.03889 | - | - | - | - |
| *F. Superiore* | -11.03888 | - | - | - | - |
| *(6) Travignolo* | -11.03871 | 947 | 2529 | N | 0.04 |

**Table 1.** Remaining mountain glaciers in the Dolomites region analysed in this study. [a] RGI ID (RGI60-) and median elevation are retrieved from Randolph Glacier Inventory v 7.0 data (RGI, 2023). [b] Glacier area from Smiraglia and DIolaiuti (2015) inventory. [c] *Marmolada* Glacier includes data averaged from all its sectors, as they are now divided but considered as a single one here, as they were in the 1980s. [d] *Fradusta* Glacier was formerly a single glacier that split into two (*F. Superiore* and *F. Inferiore*), with the latter not existing anymore.

## 3 Data and methods

In this study, aerial photographs spanning three decades (1980-2012), airborne LiDAR (2010-2014) and UAV surveys (2023) (Table 2) have been used to assess glacier change in the Dolomites region. Data processing involved four different phases. Each one is explained in a subsection.

### 3.1 Acquisition and pre-processing

All photos used have been pre-processed using Adobe Photoshop (v. 2023) to improve the exposure of heavily shadowed areas. The archive images span from the 1980s to 2012 (Table 2, Fig. S1). Digital photos from 2010 and 2012 have an optimal quality for SfM processing but lack metadata. Older analogue aerial photographs were scanned with a non-photogrammetric scanner, and therefore do not present distortion information for camera calibration.

Airborne LiDAR surveys have been performed in 2010 and 2014 using an Optech ALTM 3100 instrument mounted on a helicopter. The produced dense point clouds include areas outside of the glaciers. The Ground Control Points (GCPs) used during georeferencing were retrieved from LiDAR point clouds only, using the open-source software CloudCompare (v.2.12). Prior to



| Year - Archive Images | Date [a] | Campaign name | Size | Format | Coverage [b] |
|---|---|---|---|---|---|
| 1980 | 23-25 Jul. | *Reven Belluno* | 23 x 23 cm | A (BW) | 3 |
| 1982 | - | *Reven M. Veneta* | 23 x 23 cm | A (BW) | 4, 5, 6 |
| 1991 | 19 Sept. | *Reven M. Veneta* | 23 x 23 cm | A | 4 |
| 1992 | 8-21 Aug. | *Reven M. Veneta* | 23 x 23 cm | A | 1, 2, 3, 5, 6 |
| 1999 | 8-28 Oct. | *Reven Cadore* | 23 x 23 cm | A (BW) | 4 |
| 2001 | 14-15 Oct. | *Reven Belluno* | 23 x 23 cm | A (BW) | - |
| 2010 | 21-22 Sept. | *Reven Cadore* | 10368 x 5760 px | D | 4 |
| 2012 | 21 Aug. | *Reven Agordo* | 12983 x 8483 px | D | 5 |
| Year - Other Surveys | Date | Survey Type | Sensor | | Coverage |
| 2010 | 1 Aug. - 10 Oct. | Airborne LiDAR | Optech ALTM 3100 EA | | All |
| 2014 | 23-26 Sept. | Airborne LiDAR | Optech ALTM 3100 EA | | All |
| 2023 | 9 Sept. – 1 Oct. | UAV | DJI Mavic 2, DJI Air 2 | | 1, 2, 3, 4, 6 |
| 2023 | 10 Oct. | Aerial photos | Canon 6D MkII | | 5 |

**Table 2.** Information regarding the dataset used in this study. All the aerial photos are available online under the licence Italian Open Data License 2.0 (IODL 2.0) and are the property of *Regione del Veneto – L.R. n. 28/76 Formazione della Carta Tecnica Regionale*. 2010 and 2012 photos have been used only for visual reference and not for mass balance reconstructions. Format of archive imagery is Digital (D) or Analogue Scan (A), with some scans in black and white (BW). [a] Date related only to the portion of the flight used in this work. [b] Area IDs as for Fig. 1 represent only the successfully reconstructed scenes. The complete coverage of the aerial imagery is shown in Fig. S1.

GPCs manual identification, the PCV (*Portion de Ciel Visible*) plug-in, a generalization of Tarini et al. (2003) algorithms, was used to improve visualization. Preference was given in picking points that were steady and well recognisable over time also in lower resolution photos. The number of GCPs per area varied between 6 and 30 depending on glacier size and image quality. 2023 UAV surveys lack of GCPs and therefore needed the same use of LiDAR-derived reference points to improve accuracy and precision.

## 3.2 Structure from Motion workflow

This work uses Structure from Motion (SfM) to retrieve the former glacier surface as dense point clouds, which are later compared with LiDAR data. The SfM workflow has been entirely performed using the software Agisoft Metashape (v. 2.0.4) and is similar to a section within the work of Knuth et al. (2023) processing, where photos that have no camera calibration information available, nor fiducial markers were used. Since there was no calibration data available (i.e., non-photogrammetric scanner was used), archive images were adjusted and standardised through camera calibration. This was achieved manually by picking pseudo fiducial markers (i.e., consistent point in the photo frames notches) from each photo frame and performing the calibration in Agisoft Metashape. The photo alignment was performed using 200,000 tie points and 20,000 key points with



the maximum quality, generating a sparse point cloud as output. Georeferencing of the obtained point clouds was done using known GCPs retrieved from LiDAR. GCPs were manually placed and adjusted in all the photos.

GCPs coordinates were then imported in Agisoft Metashape, where the sparse point cloud position, scale and camera positions were updated accordingly. A triangular mesh with photo realistic texture has been computed from each dense point cloud to be used for visual interpretation and eventual dissemination of the results (Fig. 2). All the further calculations and comparisons have been done using dense point clouds. Digital Elevation Models (DEMs) and orthophotos were also generated for each survey to enable the manual mapping of glacier areas. The full report of the SfM output is available in the supplementary material (Table S1).

### 3.3 Point cloud comparison and error assessment

SfM dense point clouds have been used as primary output to detect and quantify surface elevation change. The entire processing was conducted in CloudCompare. LiDAR point clouds were used to refine alignment through manual point picking. This was necessary to adjust SfM point clouds from historical imagery only. Once aligned, point clouds from different years have been compared using the Multi Scale Model to Model Cloud Comparison (M3C2, Lague et al., 2013) algorithm. M3C2 is a robust solution to assess the differences between point clouds without the need of surface meshing or DEM generation. M3C2 workflow efficiently manages complex terrain features like flat and vertical surfaces within a single scene and was already used in similar applications (e.g., Midgley and Tonkin, 2017; Mishra et al., 2022; Securo et al., 2024a). In certain reconstructions of the older scenes (1980s), the point cloud output of solely SfM contained small void areas in correspondence with the underexposed sectors or because of snow. In those cases we used the Metashape-produced 3D triangular meshes, that were sampled and processed in CloudCompare as the other point clouds thereafter.

The M3C2 output that holds the computed differences was rasterised and imported in QGIS (v. 3.28), where areas of the glaciers were manually mapped using orthophotos and DEMs. A density value of $850 \pm 60$ kg m$^{-3}$ (Huss, 2013) was implemented to convert surface elevation change to mass balance. We then calculated zonal statistics using the previously generated glacier polygons to derive the geodetic mass balance of each glacier, using common area with regards to different years. Every comparison included 2010 LiDAR data and has never been done using two historical SfM-point clouds at a time, to reduce possible sources of error.

The error in surface elevation change measurements depends on several factors: (i) LiDAR accuracy ($E_{LiD}$), (ii) alignment error between the compared point clouds ($E_{AL}$) and (iii) error of M3C2 measurements (distance uncertainty, $E_{M3C2}$). $E_{LiD}$ has been quantified by the LiDAR surveys manufacturer comparing Helicopter LiDAR data with known points measured with differential GPS, and equals $\pm 0.12$m. In this study, our comparisons were done using relative distances; therefore, it may not be considered. $E_{AL}$ was measured using M3C2 in the frontal and lateral areas outside of the glaciers but with similar average slope values to the latter. The areas characterised by the presence of ice or where significant change (Lague et al., 2013) was detected by M3C2 have been excluded from $E_{AL}$ calculation. Evident noise or artefacts have also been excluded manually. Each individual comparison reported a different $E_{AL}$. $E_{M3C2}$ was available as a direct output of the algorithm (i.e., distance uncertainty), and considering our dataset was negligible compared to the $E_{AL}$.





**Figure 2.** Examples of the reconstructed 3D textured meshes of *Antelao Superiore, Fradusta and Marmolada Principale g*laciers during different decades. Reconstructions of *Popera, Sorapiss Occidentale, Cristallo, Antelao Inferiore, and Travignolo* Glaciers are shown in supplementary materials.





Considering the quality of our dataset, reconstruction of glaciers surface using scanned images was limited in accuracy
because: (i) a lack of photogrammetric quality scans of the negatives; (ii) the resolution and exposure of the imageries in
presence of heavily shaded zones or clean snow resulted in featureless areas; (iii) in some settings it was difficult to pick up
suitable ground control points. These sources of errors are harder to quantify. Further sources of uncertainty may be related to
the temporal coverage of aerial photography that was not uniform over all glacial bodies and the years (Table 2).

### 3.4 Weather station network

Precipitation and temperature time series from twenty automatic weather stations (AWSs) of the ARPAV network were in-
tegrated with the datasets from two AWSs managed by the autonomous province of *Bolzano*, which altogether guarantee an
adequate spatial coverage of the study area (Fig. 1). The selection of the AWSs was based both on their spatial distribution
and proximity to the six mountain areas hosting Dolomites glaciers, as well as on the availability of continuous time series of
temperature and precipitation starting at least 20 years before 2022. A linear interpolation was employed to fill temperature
gaps of 3 days or fewer, while no reconstruction was performed on precipitation data. Additionally, years with missing data
exceeding 5% of the accumulation (November to April) or ablation (June to August) season were excluded from the analysis.
This was implemented at the level of individual AWSs, ensuring the availability of data for each year after averaging across
all stations. All the time series begin between 1985 and 2001 and end between 2020 and 2022, with 77% of them spanning
over more than 30 years, and 9% spanning less than 26 years. Due to fewer AWSs deployed in the 1980s and as a consequence
of data cleaning, only 36 to 64% of the stations present temperature (23 to 64% for precipitation) data from 1985 to 1991.
However, from 1992 onwards, the data coverage increases drastically.

For each station the standardised anomaly index (SAI) (Katz and Glantz, 1986) was calculated, defined as (1):

$$SAI = \frac{x_a - x_m}{\sigma} \tag{1}$$

where $x_a$ is either the total precipitation during the accumulation season (for the precipitation SAI, Pr SAI), and the mean
temperature during the ablation season (for the temperature SAI, T SAI). $x_m$ and $\sigma$ are the mean and standard deviation
of temperature and precipitation over the reference period 1991-2020. The accumulation and ablation seasons were defined
according to local climatology. Finally, SAI values were spatially averaged, providing unique Pr and T SAI values for the entire
region. The pre-processing applied to AWS data may result in an underestimation of total precipitation ad therefore of the Pr
SAI. Conversely, with regards to temperatures, we expect that the reconstructed data do not significantly alter the mean, as they
are prone to distribute randomly both above and below the average.

Daily snow depth measurements from four stations managed by ARPAV provided snowfall information between the 1980s-
1990s and 2023, occurring above 1900 m a.s.l (Fig. 1). These measurements were obtained from both automatic and manual
snow monitoring stations. The automatic stations deploy an ultrasound sensor and collect a datum every 30 minutes, while the
manual measurements are taken once per day at 8:00 A.M. The *Fedaia* station has the longest time series, starting in 1980,
while *Ra Vales*, *Monti Alti di Ornella*, and *Col dei Baldi* began recording data in 1993, 1986, and 1987, respectively. Using



this data, we reconstructed the October to June snow depth on the ground for the most relevant years of our study (1982, 1992, 2010, 2014, 2023). Additionally, we calculated the October to June snow depth on the ground averaged over the whole time frame for each station as well as the total annual snow accumulation.

## 4 Results

The dataset used in this study allowed us to obtain the topographic surface and area changes of Dolomites glaciers from the 1980s to 2023, allowing the determination of their geodetic mass balance. Higher accuracy and precision ($E_{AL}$ 0.1-0.3 m) were obtained during the last 13 years (2010, 2014, 2023), where only airborne LiDAR and digital photos were used. Out of the 9 glaciers analysed, *Sorapiss Occidentale, Antelao, Marmolada* and *Pale di San Martino* areas were reconstructed starting from the 1980s while *Popera* and *Cristallo* reconstruction begins in the 1990s (Fig. 1).

### 4.1 Area loss

The manually mapped areas of the Dolomites glaciers show a retreat from 4.11 to 1.81 km$^2$ between the 1980s and 2023 (-56%). The loss during the last 13 years (2010-2023) equals 0.88 km$^2$ (Fig. 3), corresponding to a relative reduction of 33%. The area losses presented here are referred only to the glaciers selected in this study and as such are an underestimation of the total area loss involving all the ice bodies in the Dolomites. In 1980s and 1990s the Dolomites glaciers were larger in number,

with several of them that have now completely melted, turned into permanent ice patches without apparent ice dynamics and heavily buried by debris.

Relative area reductions are not similar across all glaciers. Where surface debris cover is abundant like in *Popera Alto*, *Travignolo* and *Sorapiss*, we observed only minor area losses, especially during the last decade. The glaciers least affected by topographic bounding are the two affected by the largest area reduction from the 1980s: *Fradusta* (-89%) and *Marmolada*

(-60%). The latter are also the only glaciers that underwent fragmentation, with the lower part of the *Fradusta* (*Fradusta Inferiore*) being the only ice body completely extinct among our dataset. All area variations are almost completely related to glacier fronts except for *Marmolada*, where even the upper portions were affected by retreat.

*Marmolada* Glacier still remains the largest glacier of the Dolomites, contributing to 66% of total area losses from 1980s and currently representing 55% of the total glacier area of the Dolomites.

### 4.2 Surface elevation change and mass balance

Average surface elevation change has been calculated for common and total glacier area (Fig. 4). Due to the impossibility of retrieving enough data for years 1999 and 2001, we considered the period from 1990s to 2010 as a unique time frame, instead of calculating the metrics at a decadal frequency. The average cumulative surface elevation change (Table 3) was calculated for three periods: 1980s with -5.21 m, 1990s-2010s with -14.09 m and 2010s with -9.31 m.

Between 1980s and 2010 the Dolomites glaciers experienced the greatest reduction in thickness in their frontal parts, except for *Popera Pensile* and *Sorapiss Occidentale*, which instead showed greater losses in their upper portions (Fig. 4). *Marmolada*,



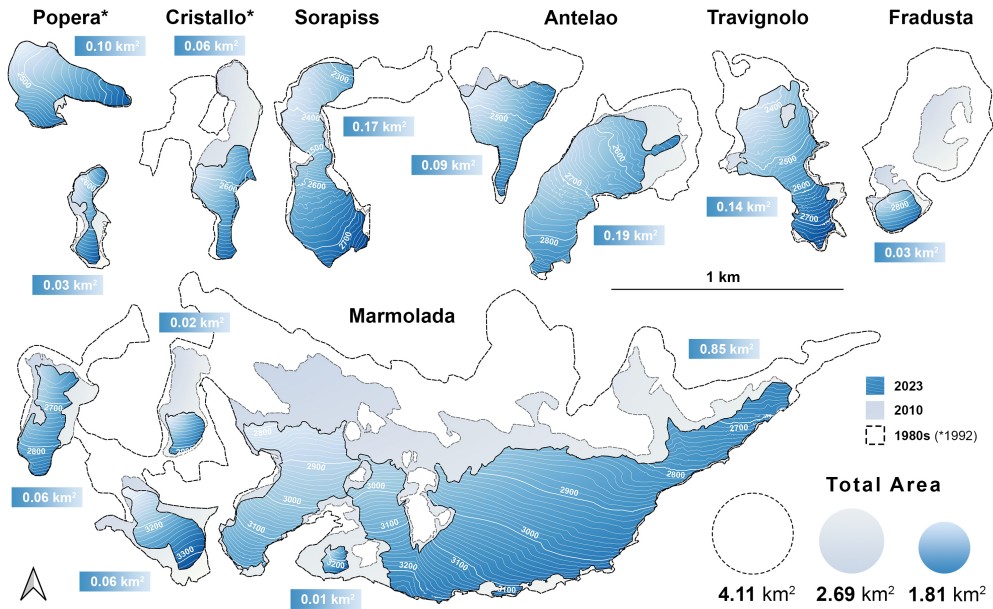

**Figure 3.** Areas of the Dolomites glaciers in 1980s (dashed line), 2010 (grey) and 2023 (blue). * 1992 for *Popera* and *Cristallo*.

*Fradusta*, and *Antelao Superiore* Glaciers showed the highest cumulative thickness losses, exceeding 50 m at their former fronts.

The largest values of average glacier surface elevation loss between 1980s and 2023 can be observed in *Fradusta* Glacier,
with an average loss of 49.70 m. The smallest changes occurred in *Popera Pensile* and *Popera Alto glaciers*, with -17.00 and -19.90 m respectively (Table 3).

Surface elevation change calculations during the 2010-2023 interval are more accurate than for the 1980s-2010 interval, with an error of 0.2-0.3 m at most, and allow a uniform comparison of the glacier behaviour during the last decade. During this time interval the acquisition are consistent with dates close to the end of the ablation season, and all glaciers were surveyed (Fig.
5a). The average surface elevation changes between 2010 and 2014 are shown in Fig. S3. *Marmolada Principale, Marmolada Ovest* and *Antelao Superiore* experienced the highest loss reaching over 30 m. The highest absolute losses, corresponding to almost 35 m, are reached in the area involved in the ice avalanche that happened in a detached part of *Marmolada Principale*, on 3rd July 2022, as shown by the Kernel Density plots of surface elevation loss (Fig. 5b). The *Fradusta Inferiore* Glacier was not included in the common area measurements as it had already disappeared before 2023 surveys took place.

During the last 13 years, we observed the only thickness increases in the central part of the *Sorapiss Occidentale* Glacier. On that glacier a rise of more than 10 m has been observed close to a wide serac whose presence is possibly related to a small surge induced by a recent rockfall (Fig. 5a) in the accumulation area as well as by internal glacier dynamics. A very small area of the *Marmolada Principale* Glacier also retains a similar surface elevation to that of 2010. This is well visible in Fig. 6a,



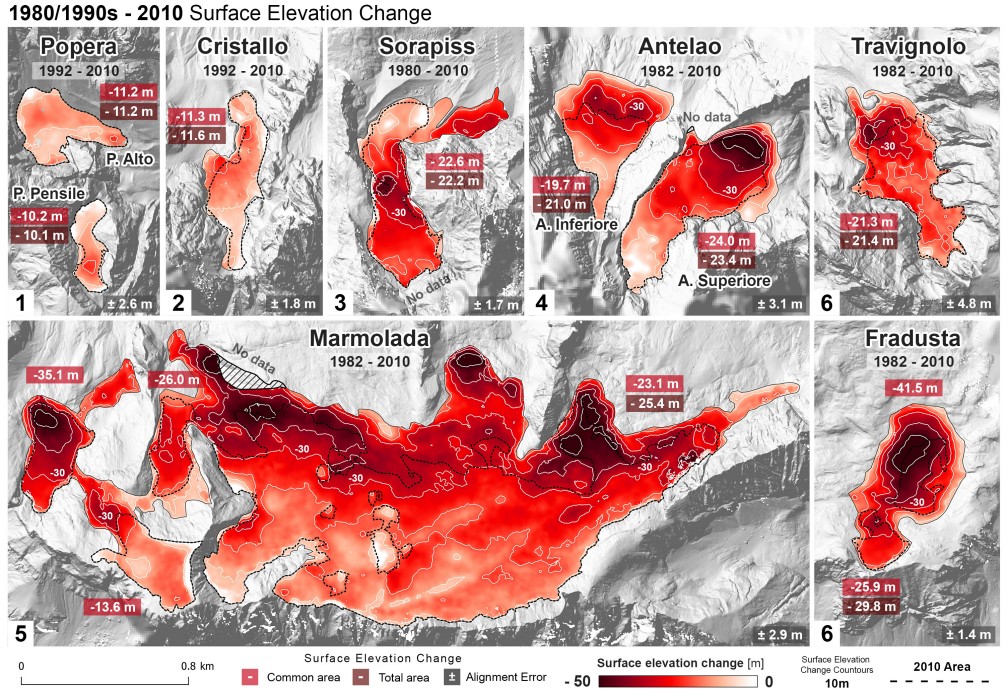

**Figure 4.** Surface elevation change (m) of the Dolomites glaciers from 1982 (1980 for *Sorapiss*; 1992 for *Popera* and *Cristallo*) to 2010. Average surface elevation change is reported for every glacier using common area (red) and total area (brown).

where it appears as an oblique line crossing the glacier from SW to NE. This feature is not natural, being related to the artificial
redistribution of snow operated to maintain a ski route on the glacier.

The average annual mass balance rate for the Dolomites glaciers varies from -0.45 ± 0.04 m w.e. yr$^{-1}$ in the 1980s, to -0.65 ± 0.05 m w.e. yr$^{-1}$ between 1990s and 2010, and -0.63 ± 0.05 m w.e. yr$^{-1}$ in the last 13 years (Table 4).

Total volume loss in the entire observation period equals approximately 0.105 Gt, with 0.083 Gt lost between the 1980s and 2010, and 0.022 Gt lost during the 2010-2023 interval. The *Marmolada* Glacier by itself counts for 65.7 % of total volume
losses. Other major contributors include *Antelao Superiore* and *Fradusta* Glaciers with 7.9 % and 7.3 %, respectively.

Our results show that the use of a fixed maximum glacier area in the geodetic mass balance leads to an underestimation of the m. w.e. loss when compared to common area calculations. In our case the bias introduced by total area is between -1% and -31% of the common area mass balance, depending on the site and considered period. There are some cases of decadal comparison (1980s-2010 in *Cristallo, Antelao Inferiore* and *Marmolada*) where total glacier area produced larger mass balance
losses than calculations using common area.

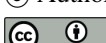



| Glacier | Common Area Surface Elevation Change (m) | | | | |
|---|---|---|---|---|---|
| | **1980s** | **1990s** | **2000s** | **2010s** | **Total** |
| | 1980-82 to 1991/92 | 1991-92 to 1999/01 | 1999-2001 to 2010 | 2010 to 2023 | All period |
| *Popera Alto* | - | -11.2 ± 2.6 | | -8.7 ± 0.2 | -19.9 |
| *Popera Pensile* | - | - 10.2 ± 2.6 | | -6.8 ± 0.2 | -17.0 |
| *Cristallo* | - | - 11.3 ± 1.8 | | - 10.1 ± 0.3 | -21.4 |
| *Sorapiss Occidentale* | -3.0 ± 1.7 | - 19.6 ± 1.4 | | - 3.4 ± 0.2 | -26.0 |
| *Antelao Superiore* | -6.6 ± 3.1 | - 4.7 ± 2.3 | - 12.7 ± 2.3 | - 7.5 ± 0.3 | -31.5 |
| *Antelao Inferiore* | - 1.4 ± 3.1 | -5.7 ± 1.7 | -8.6 ± 1.6 | -11.7 ± 0.3 | -27.4 |
| *Marmolada* [a] | -8.9 ± 2.9 | - 14.7 ± 1.8 | | -10.6 ± 0.3 | -34.2 |
| *Travignolo* [b] | -21.4 ± 4.8 | | | - 10.0 ± 0.2 | -31.4 |
| *Fradusta* [a] | -6.2 ± 1.4 | -28.5 ± 1.8 | | -15.0 ± 0.3 | -49.7 |

**Table 3.** Average surface elevation change measured on common glacier area (for each comparison) for the 9 Dolomites glaciers. 1980s reference year is 1982 (1980 for *Sorapiss*); 1990s is 1992 (1991 for *Antelao*), 2010s is 2010 and 2020s is 2023. Maps of the surface elevation change of the first period (1980s-2010) retrieved from historical SfM and the modern data (2010-2023) are presented respectively in Fig. 6 and 7. Alignment Error ($E_{AL}$) is shown for every comparison. [a] *Marmolada* and *Fradusta* sectors are treated a single glacier. [b] *Travignolo* has just one cumulative value from 1982 to 2010.

| Glacier | Mass Balance Rate (m w.e. yr⁻¹) | | | | |
|---|---|---|---|---|---|
| | **1980s** | **1990s** | **2000s** | **2010s** | **Average Rate** |
| | 1980-82 to 1991-92 | 1991-92 to 1999-01 | 1999-01 to 2010 | 2010 to 2023 | All period |
| *Popera Alto* | - | -0.31 ± 0.02 | | - 0.57 ± 0.04 | - 0.55 ± 0.04 |
| *Popera Pensile* | - | -0.28 ± 0.02 | | - 0.44 ± 0.03 | - 0.47 ± 0.04 |
| *Cristallo* | - | -0.31 ± 0.02 | | - 0.66 ± 0.05 | - 0.59 ± 0.04 |
| *Sorapiss Occidentale* [a] | - 0.21 ± 0.02 | -0.54 ± 0.04 | | - 0.35 ± 0.03 | - 0.51 ± 0.04 |
| *Antelao Superiore* | - 0.62 ± 0.05 | -0.57 ± 0.08 | - 0.98 ± 0.07 | - 0.77 ± 0.05 | - 0.65 ± 0.05 |
| *Antelao Inferiore* | - 0.13 ± 0.01 | -0.69 ± 0.05 | -0.66 ± 0.05 | - 0.49 ± 0.04 | - 0.57 ± 0.04 |
| *Marmolada* [b] | - 0.75 ± 0.06 | - 0.70 ± 0.05 | | - 0.69 ± 0.05 | - 0.71 ± 0.05 |
| *Travignolo* | - 0.65 ± 0.05 | | | - 0.65 ± 0.05 | - 0.65 ± 0.05 |
| *Fradusta* [b] | -0.52 ± 0.04 | -1.34 ± 0.10 | | - 0.73 ± 0.07 | - 1.03 ± 0.08 |
| All glaciers | -0.45 ± 0.04 | -0.65 ± 0.05 | | -0.63 ± 0.05 | -0.64 ± 0.05 |

**Table 4.** Mass Balance rates (m w.e. yr⁻¹) of the Dolomites glaciers during the last four decades. The uncertainty interval coming from the density conversion of 850 ± 60 kg m³ is reported for all measurements. (a) *Sorapiss Occidentale* values have been corrected removing the positive elevation gain portion for 2010-2023; (b) *Marmolada* and *Fradusta* mass balance are based on the average of all (now segmented) parts of the former glacier. All calculations are based on common areas.

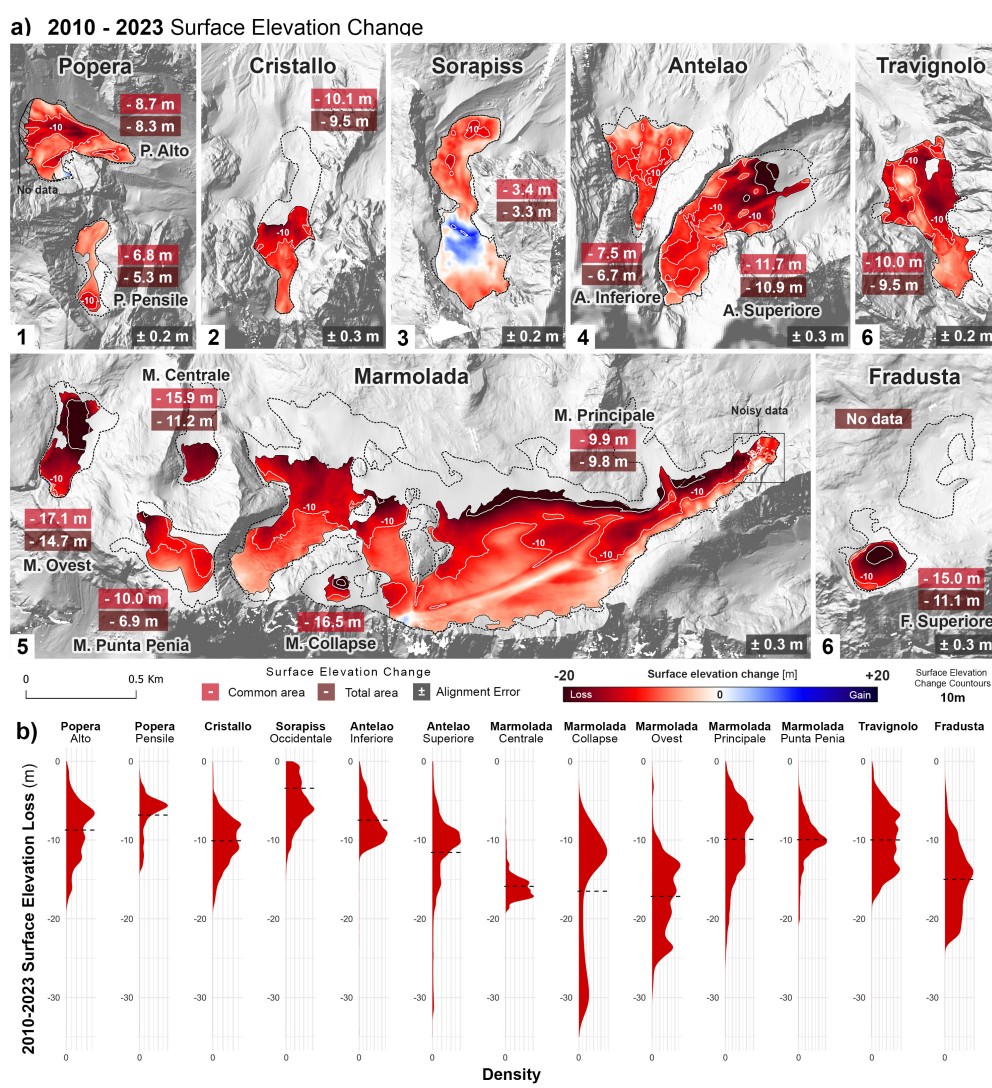

**Figure 5.** Surface elevation change (m) measured across the Dolomites glaciers from 2010 to 2023. Average surface elevation change is reported for every glacier using common area and total area (a). Kernel density distribution plot for surface elevation loss with mean loss (black dotted line), with all former *Marmolada* Glacier section presented as divided plots (b).





## 4.3    Climate data

The general trend for temperature and precipitation SAI shows more frequent positive values towards the end of the 1985-2022 time frame (Fig. 6a), meaning above long-term average temperatures and precipitation. On the contrary, negative SAI means below the long-term average temperature or precipitation. The nine lowest T SAI values fall within the first 15 years (1985-2000), with the only exception of 2014 (T SAI = -1.23). Among the ten highest events, seven have occurred in the last 15 years (2007-2022). Exceptions are represented by years 1994, 1998 and 2003, this latter showing the highest T SAI with a value of 2.74. The minimum T SAI was recorded in 1989 with -2.18. Concerning the Pr SAI, the rising trend is less pronounced than for temperature, but still shows a clear increase. Indeed p-values equal $1.1 \cdot 10^{-2}$ for Pr SAI and $1.9 \cdot 10^{-5}$ for T SAI. In this case, six of the ten lowest Pr SAI fall before 2000, and eight of the highest are recorded after 2008. The maximum Pr SAI has been calculated for 2014 with a value > 2, while 1996 is marked by the minimum value at -1.22. In the first half of the time series the T SAI tends to be lower than the Pr SAI, while in the second half, this trend progressively reverses. Furthermore, SAI values, particularly regarding temperature, have become more positive towards the end of the time frame. Temperatures have risen by 0.4-0.6 °C per decade since 1985, while precipitation showed an increase that lasted about 15 years from 1995, culminating in the extremely snowy year of 2014 (Fig. 6b). *Fedaia* station, the only one providing data since 1980, does not show any trend for the total snow accumulation (p-value = 0.61; Fig. 6c), however, increased extreme events can be observed in the last decade of its time frame. The other three snow monitoring stations exhibit slightly different patterns, demonstrating a higher frequency of snowy winters also in previous decades.

Snow height on the ground shows a peak in March or April depending on the station altitude (Fig. 6b). Among the investigated years, 2023 and 1992 values lie below the mean for most of the accumulation season, getting close to or even overtaking the mean only in April (April and May for the highest stations of *Ra Vales* and *Monti Alti di Ornella*).

## 5    Discussion

### 5.1    Climate trends impacting the Dolomites cryosphere

There are numerous studies describing the characteristics of the recent climate regime change occurring in the Alps (Huss et al., 2017; Hock et al., 2019). The main parameters influencing the response of the Alpine cryosphere are linked to the increase in summer temperatures and modifications in precipitation and snowfall patterns (Žebre et al., 2021). In the Eastern Alps mean summer temperature increased by about 2.0°C since 1979, leading to an increased ablation which results more effective also due to the extension of the ablation season (Colucci and Guglielmin, 2015; Colucci et al., 2021). Mean annual air temperature increased by 0.3 ± 0.2°C per decade. On the other hand, while the fraction of precipitation falling as rain has increased compared to that of snow in low and mid-altitude areas, this effect has not yet been observed in winter at high-altitude glacial basins. Additionally, there is generally an average increase in snowfall amounts during the winter season at high altitudes in the Alps from 1971 (Matiu et al., 2021). However, the increase in snow, particularly linked to extreme events (e.g., the winter of 2014), has not been able to counterbalance the increase in summer temperatures, resulting in consistently negative



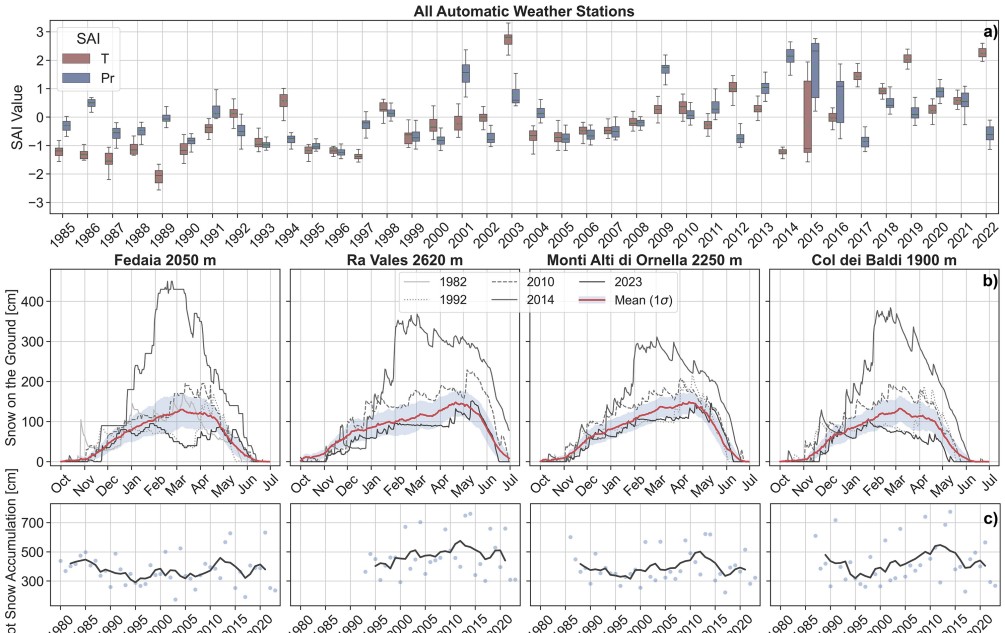

**Figure 6.** T SAI (red boxes) and Pr SAI (blue boxed) calculated between 1985 and 2022 and averaged over 22 AWS. The boxes show the quartiles (Q) of the dataset, with the central black line representing the median and green line the mean. The whiskers extend to Q1 – 1.5*(Q3-Q1) and Q3+ 1.5*(Q3-Q1), where Q1 and Q3 are the first and third quartiles (a). Snow depth on the ground between October and June for the most relevant years of the study and the 1980s/1990s – 2023 mean $\pm$ 1 $\sigma$ (red line and grey shade) for *Fedaia*, *Ra Vales*, *Monti Alti di Ornella*, and *Col dei Baldi* stations (b). Total annual snow accumulation and 5-year centered running mean (black line) for the same snow monitoring stations (c).

mass balances after 2001 (Hugonnet et al., 2021). Only in very limited areas where the highest mean annual precipitation is recorded, some small residual glacial bodies have slowed their reduction (Scotti et al., 2014; Colucci et al., 2021).

In the Dolomites, a slight increase in winter snowfall has been observed at some high-altitude stations, such as *Ra Vales* site at 2620 m (Fig. 6). Using the SAI index, it is possible to observe how in the last two decades unfavourable years conditions for glaciation prevailed. Only a few exceptions exist, notably 2014 which shows a winter precipitation anomaly with Pr SAI > 2. That year was very favourable for glaciation in all of the eastern Alps (Colucci et al., 2021), although the overall Alpine mass balance remained negative (WGMS, 2021). On the contrary, the mass loss of the glaciers was particularly noteworthy in the

hydrological year 2021/2022 (Voordendag et al., 2023). 2022 was by far the most extreme melting year among those analysed, with an extremely effective ablation season (T SAI = 2.24) and a dry accumulation season (Pr SAI = -1.58). Within Alpine mass balance records, the ablation season of 2022 results unprecedented.

     Overall, in the Dolomites we can observe an extremization of the ablation period with higher mean and extreme temperatures associated with a slight increase in snowfall on glacial basins, which is however insufficient to counterbalance the temperature



increase. According to such climatic evolution, the Dolomites are rapidly turning from being mountains hosting sites favourable
to local glaciation, to areas where peri-glacial processes will progressively gaining importance.

## 5.2    The Dolomites glaciers within the European Alps

The European Alps are home to 12 Reference Glaciers (RGs) with available glaciological mass balance data from before 1980
(WGMS, 2021). These can be used as benchmarks for our results, considering that the study area lacks any long-term mass
balance monitoring program. All the Dolomites glaciers exhibit a mass balance within the range of the RGs, with values at
the upper end, similar to the RGs that have lost less m w.e. during the last four decades (Fig. 7). The *Fradusta* Glacier is the
only site that presents a cumulative mass balance close to the 1980-2023 RG average. Glaciers with debris cover (*Popera Alto*
and *Pensile, Sorapiss Occidentale*) or topographically shadowed (*Cristallo, Antelao Inferiore*) display mass balances that are
comparable to those of other RGs (e.g. *Kesselwandferner, Allalingletscher, Giétro Glacier*), although these are larger and lie
at higher altitudes. In terms of decadal mass balance rates, the Dolomites match the behaviour of RGs during the 1980s (-0.45
m w.e. yr$^{-1}$ in average), while are always lower than the average rates during the following decades. Dolomites glaciers mass
balance rates are half of the average RGs rate during the last 13 years, with -0.63 m w.e. yr$^{-1}$ against -1.27 m w.e. yr$^{-1}$. If
we consider all 17 RGs available in Central Europe from 2010 to 2023 the average mass balance rate equals to -1.44 m yr$^{-1}$
(WGMS, 2021).

320        The apparent resistance of the Dolomite glaciers to climate change is due to several factors: (i) the orographic protection
offered by the complex topography of these mountains; (ii) the importance of avalanches in the dynamics of these glaciers;
(iii) the abundant debris cover that affects the remaining glacial volumes. All these factors contribute to dampening the effects
of climate change, as observed in similar contexts in the Alps. Glaciers behaving similarly are for example found in the
easternmost sector of the Alps (Julian Alps), or in the Orobic Alps (Southern-Central Alps), where topography feedback and
conditions similar to the ones characterizing the Dolomites glaciers are found. *Montasio Occidentale* glacier and *Canin East*
ice patches, in the Julian Alps, share similar mass balance rates, with -0.09 m w.e. yr$^{-1}$ (2006-2019; De Marco et al., 2020)
and -0.10 m w.e. yr$^{-1}$ (2010-2023; Colucci et al., 2021). *Lupo* Glacier, the only glacier in the Orobic Alps which is routinely
monitored by *Servizio Glaciologico Lombardo* (SGL), instead exhibits a mass balance rate of -1.08 m w.e. yr$^{-1}$ (2010-2023),
although heavily negatively influenced by 2022 and 2023 seasons. Other glaciers monitored by SGL like *Suretta Sud* (data
until 2022) and *Paradisin Campo Nord* show a mass balance rate of -1.30 and -1.29 m w.e. yr$^{-1}$, respectively (Scotti et al.,
2014; Hagg et al., 2017). Anther italian region of the Eastern Alps where systematic glacier monitoring is being undertaken
is South Tyrol. Here the mass balance available for 5 glaciers from 1997 to 2017 showed an average rate of -1.09 m w.e. yr$^{-1}$
(Galos et al., 2022).

        The occurrence of extremely snowy winters is able to stabilise the dynamic of some glaciers of the Dolomites. This is evident
at the end of summer of 2014, when 5 glaciers of the Dolomites (*Popera, Sorapiss, Antelao Inferiore, Marmolada*) recorded a
net positive mass balance (2010-2014). Only *Fradusta* continued thinning at a high rate because of its exposure to sunlight and
its accumulation not relying on snow avalanches at all (Fig. 7). The highest values of residual snow on glacier surface during



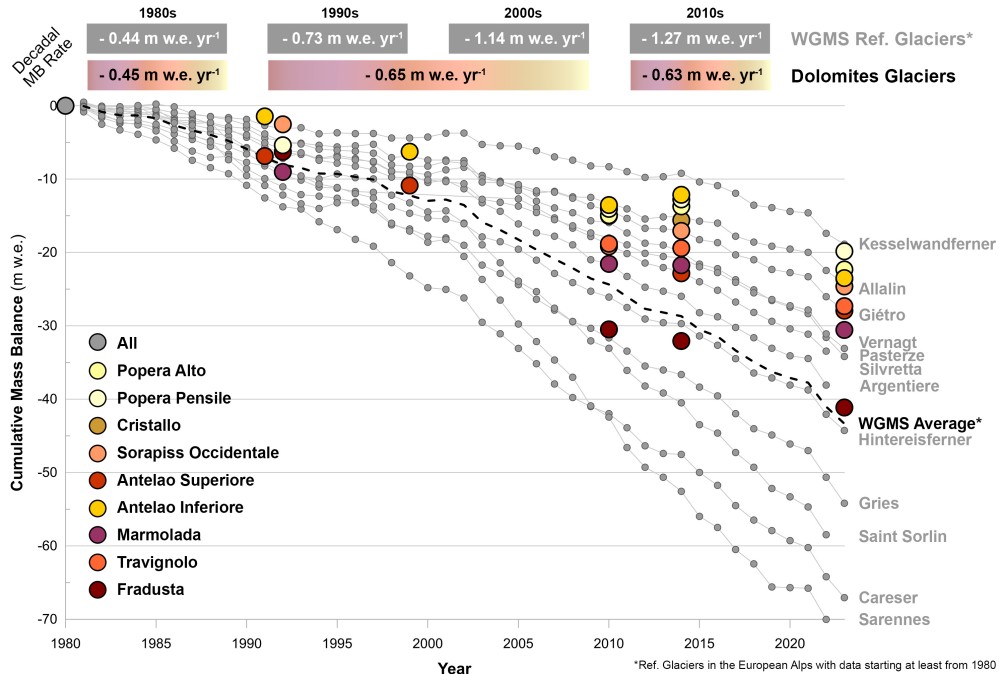

**Figure 7.** Mass balance (mb) rates of World Glacier Monitoring Service's (WGMS, 2021) RGs with *in-situ* data starting earliest in 1980 and the Dolomites glaciers. Among our mb, measurements starting from 1982 have been corrected to 1980 using the average mb loss. Mb of Dolomites glaciers with measurements starting from the 1990s have been calculated back to the 1980s using the average mb rate of all Dolomites glaciers during the 1980s (i.e., 0.45 m w.e. yr$^{-1}$). Three RGs lack 2023 data.

this shorter period are found on glaciers (e.g. *Antelao Inferiore,* upper part of *Sorapiss Occidentale, Popera Pensile and Alto*) that are prone to avalanche accumulations being surrounded by towering and steep cliffs.

Cumulative geodetic mass balance measurements derived from single years during different decades, as used in this study, can be impacted by seasonal variability influencing the surveys. The presence of residual snow on glacier surface and firn, can alter the measurements introducing a bias. This bias is possibly already included in the estimation range of $850 \pm 60$ kg m$^{-3}$ used for density conversion. From our observations, this might have influenced data in 1980s, ensuing geodetic results with a negative bias in the mass balance, albeit within the methodology error. 1991-1992 glacier conditions at the time of the

survey instead appear optimal. The same issues that affected the 1980s surveys may have influenced the 2010 surveys too, although with less intensity. Data from 2014 cannot be considered useful for assessing the state of glaciers, as thick residual snow was present on almost all the considered sites. Surveys from 2023 were carried out during the best possible time window and present no bias.




### 5.3 Local glaciological variability within the Dolomites

The glaciers of the Dolomites are all classified as very small glaciers (< 0.5 km$^2$), except for *Marmolada Principale*. Their short-term mass balance is influenced by local topography and avalanche accumulation, rather than by climate conditions alone.

An example of the influence of local topography on glacier evolution is *Sorapiss Occidentale*, where a steep rock wall separates the upper cirque from the lower portion of the glacier. The upper portion, where the highest thickness losses are observed, was completely debris free until a recent rockfall, which occurred after 2010 (Fig. 8a). The lower part of the glacier,

where a thick layer of debris is present, extends almost up to the frontal moraine. The overload caused by the recent rockfall appears to have produced a small surge in its central crevassed portion, as proven by the advanced position of the crevasses between 2010 and 2023 surveys. The increases in the surface elevation are related to the volume of the rock and debris fallen but cannot explain the relative elevation gain values larger than 10 m by itself (Fig. 8a). Similar processes, involving actual surges and not only partial advances, have been observed for other Alpine glaciers before on rare occasions (e.g., Deline, 2001).

The *Marmolada* Glacier, in its main sector, *Marmolada Principale*, is by far the largest and most representative glacier of the area. Thickness losses here appear to be inversely proportional to its elevation during all four last decades, although the maximum values measured in the recent period (2010-2023) are linked to the ice avalanche involving one of its highest sectors on July 3$^{rd}$ 2022 (Fig. 8b). The episode and its potential triggers are described in Bondesan et al. (2023) while glacier conditions in 2004 and 2014 have been previously investigated by Santin et al. (2019) and Forte et al. (2020). Forte et al.

(2020) highlighted evidence of cold ice areas in several sectors of *Marmolada*. The former *Marmolada* glacier is now currently segmented into 6 different portions, each with varying degrees of thickness losses (Fig. 5b). It should be noted that the thickness losses of the largest section of *Marmolada Principale* may have been reduced by anthropic activities related to the ski slope maintenance and the use of geotextile sheets, as reported during some ablation seasons (e.g., Baroni et al., 2015).

Mapping the exact extent of fully debris cover glacier portions like *Popera Alto* and *Sorapiss Occidentale* is not straight-

forward, especially concerning the precise locating of the glacier's front. Dead ice in front of the actual glacier are not always easy to distinguish from the main glacial body if additional data (e.g., ground penetrating radar) are not available (Santin et al., 2023). Another challenging issue regards evaluating the dynamic behaviour of these debris-covered fronts. Our data confirms that because of debris these portions may be misinterpreted in mapping efforts based only on remote sensing (e.g., Fig. 9a, c). In this study we used the surface lowering observed during the last 13 years and direct observations on site to assess the

glaciers end.

Some Dolomites glaciers, such as *Fradusta Superiore*, *Popera Pensile*, and *Antelao Inferiore*, are now at the border between the classification of mountain glaciers and permanent ice patches. Despite this, they can still be classified as mountain glaciers due to the evidence of recent dynamics and internal deformation through the presence of open crevasses. For example, *Antelao Inferiore* still presents visible crevasses and *Popera Pensile* and *Fradusta Superiore* show the presence of a *bergschrund* and

annual layering (Fig. 9a, b). The outcropping ice/firn surface is often characterised by irregularly spaced convex bands of ice rich in fine sediment. These features are consistent with variations in ice/firn velocity across the glacier, with the fastest flow





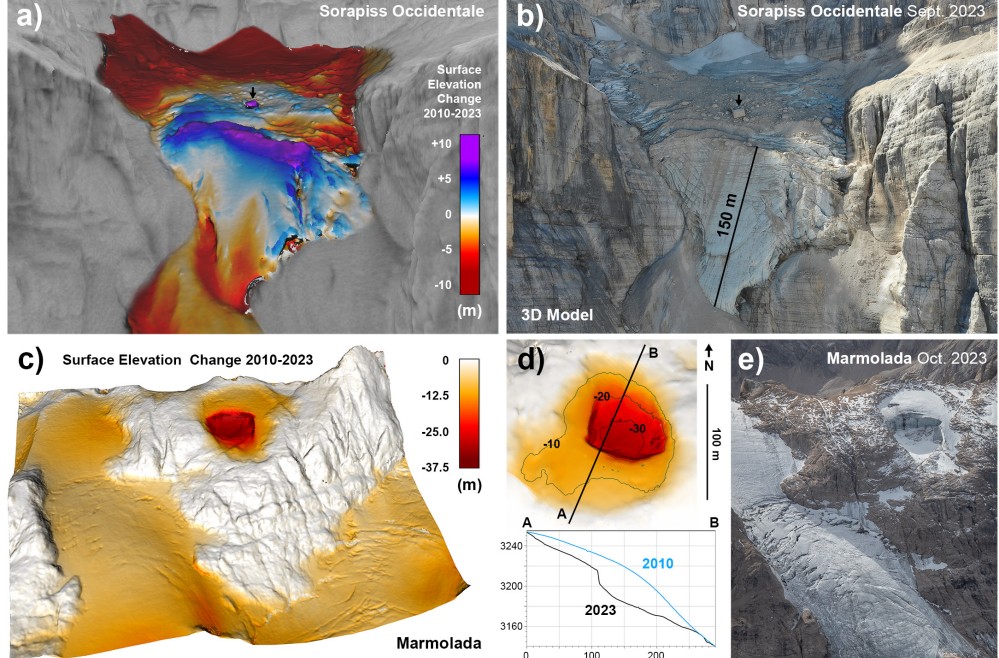

**Figure 8.** *Sorapiss Occidentale* surface elevation change (a) and 3D model (b). The black arrow is used as a reference between subplot a and b. Surface elevation change of *Marmolada* Glacier (c); zoom on the area interested by the ice avalanche happened on July 3rd, 2022 with vertical profile in 2010 and 2023 (d), and an aerial photo of the release zone (e).

being observed away from the ice margins (Hughes, 2007; Colucci and Guglielmin, 2015), suggesting the presence of ice movement.

Projecting our mass balance trends into the future is not straightforward due to the high range of morphological differences

of each ice body considered here. Processes like debris cover increase or glacier segmentation may contribute to different responses to ongoing warming. Some glacial bodies could eventually undergo a partial or complete transition from glacial to periglacial landforms, as already discussed in (Seppi et al., 2014) for the former *Cima d'Uomo* Glacier. The shrinkage and rapid disappearance of glaciers in the Dolomites may find a counterpart behaviour exhibited since the Little Ice Age in glaciers of the Julian Alps (e.g.,*Canin* Glacier and *Triglav* Glacier, Colucci, 2016; Colucci and Žebre, 2016) or in other minor glacial

bodies in *Prokletije* (Albania), *Durmitor* (Montenegro) and *Pirin* (Bulgaria) mountain ranges (Hughes, 2010; Gachev et al., 2016; Hughes and Woodward, 2016; Hughes, 2018). A similar evolution awaits all glaciers sharing low elevation and small size in the European Alps (Cook et al., 2023).



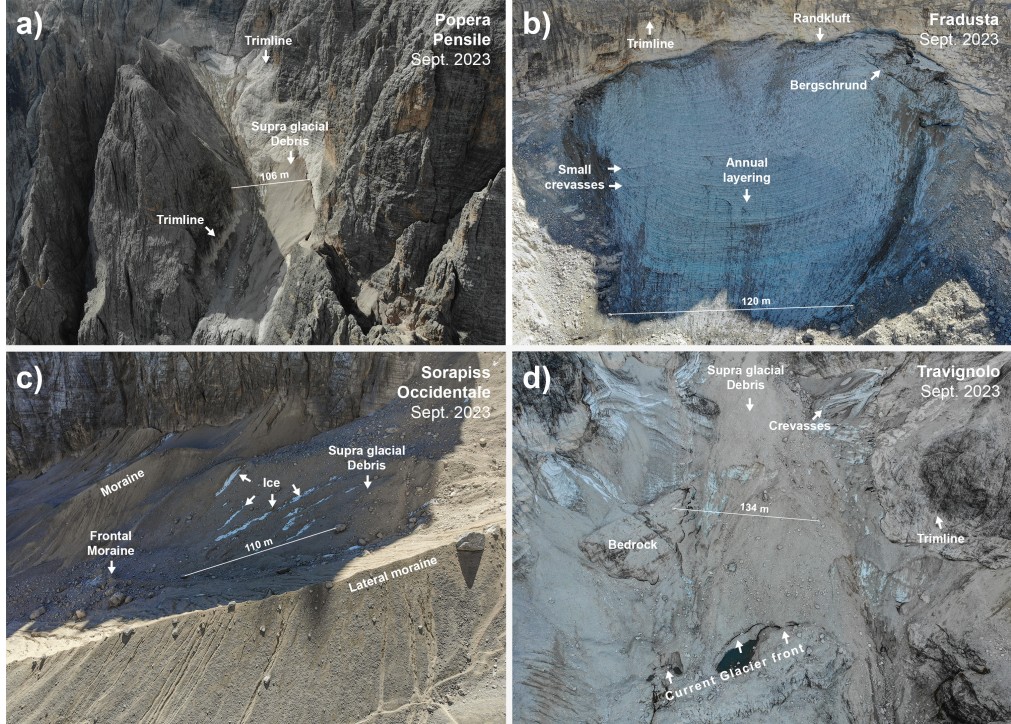

**Figure 9.** Photos of different settings of the Dolomites glaciers and their relationship with local topography: (a) *Popera Pensile*, (b) *Fradusta Superiore* (merge of multiple photos), (c) frontal part of *Sorapiss Occidentale*, (d) *Travignolo* Glacier front.

## 6 Conclusions

The last 40 years of ice melting in the Dolomites region have been reconstructed, highlighting how local topography can act as

positive feedback in slowing mass loss compared to the larger reference glaciers of the European Alps. The existing glaciers in the region lost 56% of their area and $0.64 \pm 0.05$ m w.e. yr$^{-1}$ from the early 1980s to 2023. The total volume loss equals 0.105 Gt, of which 0.022 Gt in the last 13 years (2010-2023). In the late 1950's the Dolomites were hosting 33 glaciers, of which only 9 are still active; 2 of these have also segmented into minor portions.

All the glaciers in the Dolomites currently have their accumulation areas below the envELA and are projected to disappear

or segment into minor glacial bodies in a few decades. Their fate appears inevitable even if a 1991-2020 steady-state climate is assumed. Only locally, favorable feedback may be brought by the debris coverage. A few glacial bodies may eventually shift from glacial to periglacial, thus becoming more resilient in a warming climate.

Although this work adds new quantitative information that fills part of the existing gap, we strongly encourage more specific monitoring for some if not all these sites at least on an annual basis. This is of even greater significance in the context of the

present climate trend, which is leading to a rapid degradation of the Dolomites glacial bodies. As evidenced by e.g. the ice



avalanche of *Marmolada* Glacier in July 2022, the final stages of the disappearance of these glaciers may result in parossistic events, which in turn pose risks in an extremely popular tourist area.

*Data availability.* Archivial imagery are accessible online (https://idt2.regione.veneto.it/portfolio/aereofototeca/). LiDAR and AWSs data can be requested to ARPAV (https://www.arpa.veneto.it). 2023 surveys will be made available on request. Processed data will be made
accessible in Zenodo (Securo et al., 2024b).

*Author contributions.* Conceptualization: AS. Investigation: AS. Methodology: AS, CDG, RRC. Supervision: RRC. Visualization: AS, CDG. Writing – original draft preparation: AS, CDG, RRC, GB. Writing – review and editing: all authors.

*Competing interests.* The contact author has declared that none of the authors has any competing interests.

*Acknowledgements.* The authors would like to acknowledge ARPA Veneto and Fabrizio Tagliavini for the availability of the LiDAR surveys
campaign of the Dolomites glaciers and their weather station dataset; the Alpine-Adriatic Meteorological Society for financing the 2023 Marmolada survey; Justin Perry for the English editing of the final version of the manuscript; Daniele Fontana for the support and help in the survey campaign of 2023; Gerry De Zolt for the logistic support during the flight above Marmolada Glacier in 2023; Marco Basso Bondini, Luca Baggio, Samanta Miotti and Joao Gomez Ilha for the help during fieldwork.



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
