# Peer review of "The Glaciers of the Dolomites: last 40 years of melting"

_EGUsphere, 2024_

## Referee Comment (RC1)

**The Glaciers of the Dolomites: last 40 years of melting - Securo et al., 2024**

The authors present a comprehensive overview of the state of glaciers in the Dolomites including detailed assessments of glacier volume and mass change for the main remaining glaciers in the region. They provide an informative discussion of the observed glaciological changes in the context of local climate and topography and contrast the evolution of glaciers in the Dolomites with glaciers in other European regions. As the authors point out, glacier change data from the Dolomites is sparse (by Alps standards) and this work is a valuable contribution towards a quantitative understanding of glacier change in this region over the last ~40 years. The authors highlight the importance of local, mostly topographic factors for the evolution of very small glaciers and the need for continued monitoring to better understand the possible future trajectories of these features. I agree with this. The 3D visuals are very cool and will be valuable assets in outreach activities.

I have some questions and comments which I hope can be addressed to improve the overall clarity of the paper. My impression is that the authors probably have everything needed to do this and it is a matter of providing additional explanations or changing the way some things are presented, rather than adding to or changing the analyses. My main points are below, the following brief notes are mostly just small quibbles I had while reading. I feel like some editing for more concise language would be beneficial but this is of course somewhat subjective.

**Main questions/comments:**
**Surface change computation and treatment of errors**
The section in the methods dealing with this is a bit fuzzy and I find it hard to follow at times. The error in surface elevation change is stated to depend on lidar accuracy, alignment between the point clouds, and a distance uncertainty. The lidar accuracy is "not considered" (L163) because "relative distances" are used. I am unsure what the reasoning for this is. If I understood this section correctly the authors are comparing all other data to the 2010 lidar (L158), but this does not appear to explain why the vertical accuracy of the lidar is not a relevant factor (?) Are you only looking at the horizontal accuracy? If so, why?

I would also like more explanation of the process mentioned in L158: *"Every comparison included 2010 LiDAR data and has never been done using two historical SfM-point clouds at a time, to reduce possible sources of error."* The results show surface elevation change values for various time steps before and after 2010. How were these generated if everything is compared to 2010? I do not see how comparing everything to 2010 first and then computing differences for other periods would cancel out the errors in the historical point clouds. I may have misunderstood what you did here but either way I think it requires some more explanation. Perhaps some sort of diagram showing the processing steps to arrive at surface elevation change for different time periods would help, or just a more structured explanation.

In the results (L206) it is stated that "*Higher accuracy and precision (E_AL 0.1-0.3 m) were obtained*" for the more recent years. Since the alignment error is considered the main source of uncertainty (this is my understanding based on the methods section) it would be interesting to

see these values for the historical data as well and to include some more information on how this error was determined. Can you quantify the total error of the volume change data for the different time steps?

**Glacier area**
The authors repeatedly refer to "common area" vs. "total area". It is not entirely clear to me what they mean by common area and how it differs from total area, nor did it become clear to me which year (or average) they used for computations of volume and mass change. It would be beneficial to have a clear explanation especially of "common area" early on in the manuscript (methods section). A bias related to usage of different areas is discussed later on and it is apparent that the authors are aware of the influence of glacier area on further computations, so I think i is again just a matter of improving the clarity of how this is presented.
This publication may be of interest:
Florentine C, Sass L, McNeil C, Baker E, O'Neel S. How to handle glacier area change in geodetic mass balance. Journal of Glaciology. Published online 2023:1-7. doi:10.1017/jog.2023.86

Note that the glacier area is also related to overall uncertainties. The uncertainty in volume change is a function of the uncertainty in surface elevation change and uncertainty in the area. Neither of these factors seem to be included in the uncertainty estimate for the mass change given in the results, which appears to be based only on the uncertainty of the density conversion. I understand that it may not be possible to fully quantify the uncertainty but it would be good to at least mention this and explain that challenges related to exact area delineation (which you mention) also affect volume and mass change estimates. Note the large impact of area on uncertainties shown e.g. in Hugonnet et al 2021 (extended data Fig 5, https://www.nature.com/articles/s41586-021-03436-z).

**Abstract**
*L 8, L 50 and elsewhere in the manuscript:  unmanned aerial vehicle*
Please consider using the neutral term "uncrewed aerial vehicle"

*L 10: from 1980s to 2023*
The 1980s and 1990s are frequently referred to as time periods throughout the text. I feel like more specific phrasing would be helpful for the reader. In the abstract and as you explain your workflow it would be good to know that, e.g. "1980s" refers to data from 1980 or 1982 as per table 2.

*L10: …33% of which between 2010-2023…*
Missing word? → of which occurred (?) between 2010-2023

*L11  negative with a smaller amplitude*
Consider changing to "less negative" for clarity

**Introduction**
*29: valley bottoms*
I think "valley floors" is the more common term for this

*L62: providing a description of the glaciers in the Dolomites that are still active,*
How do you define "active" glaciers?

**Previous studies**
*L69  No glacier in the area has mid or long-term mass balance dataset available*
Missing "a"? (has a mid or long-term mass balance dataset…)

*L88 Results show an area variation of approximately -50% from 1910 to 2009.*
Consider rephrasing for clarity: "...show an area loss of …"

L91 and following
Consider restructuring for clarity. You could move the sentence starting with "also of great significance" to the end of the paragraph so that the sentence citing Serrano et al (2021) appears directly after the first use of the term ice patch.
Why is the debris cover of great significance? You might state that it is abundant without using the word significance, which is often associated with statistical parameters.

*L106  Other Dolomites massifs that still host minor ice deposits devoid of any evidence of dynamics are not included in this work*
What do you consider evidence of dynamics and how did you determine that none is present at these features compared to the nine you study?

Table 1: state in the caption or in the table for which year the area value is valid. Same year as the cited publication?
Caption: Smiraglia and Dlolaiuti → typo

**Data and methods**
Table 2: *2010 and 2012 photos have been used only for visual reference and not for mass balance reconstructions*
Would this be an opportunity to compare results using the 2010 photo vs. the 2010 lidar and assess the difference in elevation change between the different methods/data sources?

L157: *…using common area with regards to different years.*
Unsure how to interpret this - does this mean you used the same area value for all computations of geodetic mass balance? Which area value (from which year) did you use?

L157: *Every comparison included 2010 LiDAR data and has never been done using two historical SfM-point clouds at a time, to reduce possible sources of error. ..  Every comparison included 2010 LiDAR data and has never been done using two historical SfM-point clouds at a time, to reduce possible sources of error.*

Does this mean you compared every other year to 2010? See general comment above.

L163 *±0.12m* I am assuming this refers to vertical accuracy? Consider clarifying
*In this study, our comparisons were done using relative distances; therefore, it may not be considered*
I don't understand what you mean here. Are you saying uncertainties in the lidar measurements are not considered? Please clarify why not.

*EM3C2 was available as a direct output of the algorithm (i.e., distance uncertainty), and considering our dataset was negligible compared to the EAL.*
So E_al was the main error source? Can you quantify the relative contribution of the different errors?

L170 *imageries* → imagery

**Weather station network**

L180
*Additionally, years with missing data exceeding 5% of the accumulation (November to April) or ablation (June to August) season*
Unusual definition of accumulation and ablation season, please explain the reasoning behind this. What happens in the missing months? (May, September, October)

L182 *This was implemented at the level of individual AWSs, ensuring the availability of data for each year after averaging across all stations*
Why average over all of them? If the goal is to get one T&P time series for the region, consider leading with that.

L183 *All the time series begin between 1985 and 2001 and end between 2020 and 2022*
Does this mean that none of the time series extend beyond 2022?

L189 *where xa is either the total precipitation during the accumulation season (for the precipitation SAI, Pr SAI), and the mean…*
Should this be "or the mean" ?

L191: *The accumulation and ablation seasons were defined according to local climatology*
Please specify what this means.

L191: *Finally, SAI values were spatially averaged, providing unique Pr and T SAI values for the entire region*
Does "spatially averaged" mean you produced some kind of gridded data set or is this simply one averaged time series over all the weather station data? Please clarify

L193: *The pre-processing applied to AWS data may result in an underestimation of total precipitation and therefore of the Pr SAI.*
Why? What part of the preprocessing leads to an underestimation?

L198  *collect a datum*
Consider rephrasing → record a value

L200: *Using this data, we reconstructed the October to June snow depth on the ground for the most relevant years of our study (1982, 1992, 2010, 2014, 2023).*
Reconstructed as in you averaged over October to June for the given years? Or does the reconstruction involve something more complex?

L202: *Additionally, we calculated the October to June snow depth on the ground averaged over the whole time frame for each station as well as the total annual snow accumulation.*
Could you explain your reasoning for using October to June average snow depth? Wouldn't the snow depth at the end of the accumulation season (late spring) be a more relevant metric?

**Results**
L206: *Higher accuracy and precision (EAL 0.1-0.3 m) were obtained…*
What does the EAL 0.1-0.3 m value represent? (accuracy or precision? Which years? What are the values in the years where lower accuracy(?) was obtained?)

L207  *Out of the 9 glaciers analysed, Sorapiss Occidentale, Antelao, Marmolada and Pale di San Martino areas were reconstructed starting from the 1980s while Popera and Cristallo reconstruction begins in the 1990s*
State the exact years, 1980s and 1990s is vague

L214:
*In 1980s and 1990s the Dolomites glaciers were larger in number, with several of them that have now completely melted, turned into permanent ice patches without apparent ice dynamics and heavily buried by debris.*
Consider rephrasing for clarity. Something like: In the 1980s and 1990s, there were more glaciers in the Dolomites, some of which have completely melted or turned into debris covered permanent ice patches without apparent ice dynamics.

L217 *Relative area reductions are not similar across all glaciers*
State min max range of area reduction to show variation?

L219: *topographic bounding*
Consider explaining this term

L226: *for common and total glacier area*
Please explain what you mean by common and total area. Is this stated somewhere?

L226:
*Due to the impossibility of retrieving enough data for years 1999 and 2001, we considered the period from 1990s to 2010 as a unique time frame, instead of calculating the metrics at a decadal frequency. The average cumulative surface elevation change (Table 3) was calculated for three periods: 1980s with -5.21 m, 1990s-2010s with -14.09 m and 2010s with -9.31 m.*

Does "unique time frame" just mean you used a longer time step? I think rephrasing would help clarify this, something like: *The average cumulative surface elevation change (Table 3) was calculated for three periods: 1980s with -5.21 m, 1990s-2010s with -14.09 m and 2010s with -9.31 m. Due to lack of data in 1999 and 2001 it was not possible to resolve the 1990s-2010 period at decadal frequency.*

L241 *The highest absolute losses, corresponding to almost 35 m, are reached in the area involved in the ice avalanche that happened in a detached part of Marmolada Principale, on 3rd July 2022, as shown by the Kernel Density plots of surface elevation loss (Fig. 5b)*
Can you mark this in the figure? I am unsure where I can see this in Fig 5b.

L243 *The Fradusta Inferiore Glacier was not included in the common area measurements as it had already disappeared before 2023 surveys took place.*
Again, what exactly is common area?

L246 *On that glacier a rise of more than 10 m has been observed close to a wide serac whose presence is possibly related to a small surge induced by a recent rockfall (Fig. 5a) in the accumulation area as well as by internal glacier dynamics*
Interesting! If possible consider marking this feature in the figure

L248 *This is well visible in Fig. 6a,*
Should this be Fig 5a?

L251 ff and Table 4: Do these uncertainties refer only to the uncertainty originating from the density conversion, or does this also include uncertainties in area and volume?

L256 *Our results show that the use of a fixed maximum glacier area in the geodetic mass balance leads to an underestimation of the m. w.e. loss when compared to common area calculations. In our case the bias introduced by total area is between -1% and -31% of the common area mass balance, depending on the site and considered period. There are some cases of decadal comparison (1980s-2010 in Cristallo, Antelao Inferiore and Marmolada) where total glacier area produced larger mass balance losses than calculations using common area*

I am still unsure about the differences between "fixed maximum glacier area" (this term is used for the first time here), common area, and total area.

Table 4: (a) *Sorapiss Occidentale values have been corrected removing the positive elevation gain portion for 2010-2023*

Why did this need to be corrected? Did you simply delete all positive values or was there some other correction? You measured the positive elevation change and suggested that this was due to a rockfall/surge process - what is the argument for removing the elevation gain when that is what your analysis shows?

**Climate data**

L266 . *Among the ten highest events, seven have occurred in the last 15 years (2007-2022).*
Consider rephrasing for clarity? highest → warmest

L269
*The maximum Pr SAI has been calculated for 2014 with a value > 2, while 1996 is marked by the minimum value at -1.22.*
Consider rephrasing for clarity, e.g.: Pr SAI was greatest in 2014 with 2.x and lowest in 1996 with -1.22.

L272 Temperatures have risen by 0.4-0.6 °C per decade since 1985, while precipitation showed an increase that lasted about 15 years from 1995, culminating in the extremely snowy year of 2014 (Fig. 6b). Fedaia station, the only one providing data since 1980, does not show any trend for the total snow accumulation (p-value = 0.61; Fig. 6c), however, increased extreme events can be observed in the last decade of its time frame. The other three snow monitoring stations exhibit slightly different patterns, demonstrating a higher frequency of snowy winters also in previous decades.

Did you also look at station variability for T and P? How do you identify extreme events in the snow time series?

Fig 6b: The dotted line is hard to see. I'm assuming the lines refer to hydrological year, i.e. 2023 refers to the 2022/23 winter season. Consider stating this in the caption or legend.

**Discussion**

L295 *In the Dolomites, a slight increase in winter snowfall has been observed at some high-altitude stations, such as Ra Vales site at 2620 m (Fig. 6)*
How do you determine this increase? It is not really obvious from Fig 6c and there is no mention of this in the results.

L296 *unfavourable years conditions for glaciation prevailed*
Extra word? Delete "years"

L302 *Within Alpine mass balance records, the ablation season of 2022 results unprecedented.*
Missing word? (...results were unprecedented…)

L305 *According to such climatic evolution, the Dolomites are rapidly turning from being mountains hosting sites favourable to local glaciation, to areas where peri-glacial processes will progressively gaining importance.*
→ gain importance

L317 *Dolomites glaciers mass balance rates are half of the average RGs rate during the last 13 years*
Interesting!

L334  *stabilise the dynamic of some glaciers of the Dolomites*
Do you actually mean dynamic as in movement or something else? Consider rephrasing

Fig 7: Cool figure! I'd be interested in seeing how the WGMS annual product compares to your values for the Dolomiti glaciers (just an idea, the figure is informative as is and this is not needed for the manuscript)
https://cds.climate.copernicus.eu/cdsapp#!/dataset/derived-gridded-glacier-mass-change?tab=overview

L360 *most representative*
If it is the largest it is not the most representative in terms of size. Consider removing this.

L374 *In this study we used the surface lowering observed during the last 13 years and direct observations on site to assess the glaciers end*
I would like to read this earlier, eg in the methods.

L397 *In the late 1950's the Dolomites were hosting 33 glaciers, of which only 9 are still active;*
Define somewhere what you mean by active

L402 *A few glacial bodies may eventually shift from glacial to periglacial, thus becoming more resilient in a warming climate.*
There seems to be an ongoing discussion about how and whether glacial features can turn into periglacial features (e.g. discussion comments here:
https://tc.copernicus.org/articles/18/1669/2024/tc-18-1669-2024-discussion.html)
Perhaps rephrase this sentence to avoid ambiguity. You could focus on the processes that would make the ice features more resilient without classifying them as glacial or periglacial.

---

## Author Comment (AC1)

We have replied to the comments in blue.

**The Glaciers of the Dolomites: last 40 years of melting - Securo et al., 2024**

The authors present a comprehensive overview of the state of glaciers in the Dolomites including detailed
assessments of glacier volume and mass change for the main remaining glaciers in the region. They provide an
informative discussion of the observed glaciological changes in the context of local climate and topography and
contrast the evolution of glaciers in the Dolomites with glaciers in other European regions. As the authors point
out, glacier change data from the Dolomites is sparse (by Alps standards) and this work is a valuable contribution
towards a quantitative understanding of glacier change in this region over the last ~40 years.

The authors highlight the importance of local, mostly topographic factors for the evolution of very small glaciers
and the need for continued monitoring to better understand the possible future trajectories of these features. I
agree with this. The 3D visuals are very cool and will be valuable assets in outreach activities.

I have some questions and comments which I hope can be addressed to improve the overall clarity of the paper.
My impression is that the authors probably have everything needed to do this and it is a matter of providing
additional explanations or changing the way some things are presented, rather than adding to or changing the
analyses. My main points are below, the following brief notes are mostly just small quibbles I had while reading.
I feel like some editing for more concise language would be beneficial but this is of course somewhat subjective.

**Main questions/comments:**

**Surface change computation and treatment of errors**

The section in the methods dealing with this is a bit fuzzy and I find it hard to follow at times. The error in surface
elevation change is stated to depend on lidar accuracy, alignment between the point clouds, and a distance
uncertainty. The lidar accuracy is "not considered" (L163) because "relative distances" are used. I am unsure what
the reasoning for this is. If I understood this section correctly the authors are comparing all other data to the
2010 lidar (L158), but this does not appear to explain why the vertical accuracy of the lidar is not a relevant factor
(?) Are you only looking at the horizontal accuracy? If so, why?

It's correct that we are comparing all the data to 2010 LiDAR because is the best available dataset so far. Although
it is of a much smaller magnitude than the alignment error, at least in older reconstructions, it is necessary as
suggested that all errors and their propagation are considered. In the revised version we will therefore consider
all errors and how they combine into overall accuracy: alignment error from the point clouds manual alignment,
lidar error from the surveys used as ground control points source and distance uncertainty coming from the M3C2
measurements.

I would also like more explanation of the process mentioned in L158: "Every comparison included 2010 LiDAR
data and has never been done using two historical SfM-point clouds at a time, to reduce possible sources of
error." The results show surface elevation change values for various time steps before and after 2010. How were
these generated if everything is compared to 2010? I do not see how comparing everything to 2010 first and then
computing differences for other periods would cancel out the errors in the historical point clouds. I may have
misunderstood what you did here but either way I think it requires some more explanation.

The comparisons to 2010 LiDAR only are done to have the most reliable source of data used for alignment
estimation. Error is calculated outside of the glacierized area so it's more robust if we use the best dataset as
reference. The subsequent calculation for different timesteps are done just by subtraction.

Perhaps some sort of diagram showing the processing steps to arrive at surface elevation change for different
time periods would help, or just a more structured explanation.

We had initially prepared a diagram showing all the processing steps which we have lately chosen to remove.
Looking at the methodology now and considering both reviewer comments we think that it's better to integrate
that image in the supplementary materials. See Fig. R1. Considering the current manuscript length and the
suggestion coming from Anonymous Referee #2 to shorten it, we think that Fig. R1 should not be in the main
manuscript.

[Figure]

**Figure R1**. Summary of the data processing proposed to be integrated in the Supplementary Materials.

In the results (L206) it is stated that "Higher accuracy and precision (E_AL 0.1-0.3 m) were obtained" for the more
recent years. Since the alignment error is considered the main source of uncertainty (this is my understanding
based on the methods section) it would be interesting to see these values for the historical data as well and to
include some more information on how this error was determined. Can you quantify the total error of the volume
change data for the different time steps?

Higher accuracy and precision are reported for LiDAR-to-LiDAR comparisons (i.e., 2010-2014) and for UAV-to-
LiDAR comparisons (i.e., 2010-2023) which are based on more robust data. The alignment error is visible for all
comparisons in Figure 4-5. Error treatment will be implemented, see answers to comments above.

The alignment error between historical-only (i.e., analogue based) comparisons are higher, but are not included
in our calculations as we compared everything with 2010 (see comment above)

**Glacier area**

The authors repeatedly refer to "common area" vs. "total area". It is not entirely clear to me what they mean by
common area and how it differs from total area, nor did it become clear to me which year (or average) they used
for computations of volume and mass change. It would be beneficial to have a clear explanation especially of
"common area" early on in the manuscript (methods section). A bias related to usage of different areas is
discussed later on and it is apparent that the authors are aware of the influence of glacier area on further
computations, so I think i is again just a matter of improving the clarity of how this is presented.

This publication may be of interest: Florentine C, Sass L, McNeil C, Baker E, O'Neel S. How to handle glacier area
change in geodetic mass balance. Journal of Glaciology. Published online 2023:1-7. doi:10.1017/jog.2023.86

We refer to common area as the area in common between the two timesteps considered in each comparison;
while total area is what in Florentine et al. (2023) is defined as fixed maximum glacier area. i.e., the oldest year
area of each comparison. We agree with the reviewer that this needs to be specified clearly in the text adding
also the proposed reference. As pointed out in Florentine et al. (2023) using temporally resolved areas in geodetic
mass balance studies is more robust, as we have done for each single comparison. Unfortunately, we cannot add
intermediate timesteps between the one already in use in this study because of the lack of data in between

Note that the glacier area is also related to overall uncertainties. The uncertainty in volume change is a function
of the uncertainty in surface elevation change and uncertainty in the area. Neither of these factors seem to be
included in the uncertainty estimate for the mass change given in the results, which appears to be based only on
the uncertainty of the density conversion. I understand that it may not be possible to fully quantify the
uncertainty, but it would be good to at least mention this and explain that challenges related to exact area
delineation (which you mention) also affect volume and mass change estimates. Note the large impact of area
on uncertainties shown e.g. in Hugonnet et al 2021. (extended data Fig 5).

We will provide a more solid error evaluation in the revised version of the manuscript, considering the error
propagation and the total error also in the mass balance calculation. As per the current version it is true that
the only density conversion factor is considered as source of error for mass balance calculation.

The impact of area on uncertainties is big in such large-scale studies as Hugonnet et al. (2021) but in this
specific case is less impactful as we have higher resolution data and, except for debris cover, we can map with
precision the area of each glacier. We anyway agree that this should be specified better in the discussion

**Abstract**

L 8, L 50 and elsewhere in the manuscript: unmanned aerial vehicle
Please consider using the neutral term "uncrewed aerial vehicle"

Agree and will update it, thank you

L 10: from 1980s to 2023
The 1980s and 1990s are frequently referred to as time periods throughout the text. I feel like more specific
phrasing would be helpful for the reader. In the abstract and as you explain your workflow it would be good to
know that, e.g. "1980s" refers to data from 1980 or 1982 as per table 2.

We agree with the proposed change, and we will integrate it with more specific periods whenever they are
mentioned in the manuscript.

L10: …33% of which between 2010-2023…
Missing word? → of which occurred (?) between 2010-2023

Yes, it was a mistake, thank you

L11 negative with a smaller amplitude
Consider changing to "less negative" for clarity

Agree, we will update this throughout the text

**Introduction**

29: valley bottoms
I think "valley floors" is the more common term for this

Agree, it will be changed to "valley floors"

L62: providing a description of the glaciers in the Dolomites that are still active,
How do you define "active" glaciers?

Which glaciers are active in this case is taken from the last available inventory from Smiraglia et al. (2015), as
mentioned in the text. e.g. L 104-109 and Fig. 1 caption

**Previous studies**

L69 No glacier in the area has mid or long-term mass balance dataset available
Missing "a"? (has a mid or long-term mass balance dataset…)

Correct, our mistake

L88 Results show an area variation of approximately -50% from 1910 to 2009.
Consider rephrasing for clarity: "…show an area loss of …"

We will rephrase this and other similar sentences throughout the text

L91 and following
Consider restructuring for clarity. You could move the sentence starting with "also of great significance" to the
end of the paragraph so that the sentence citing Serrano et al (2021) appears directly after the first use of the
term ice patch. Why is the debris cover of great significance? You might state that it is abundant without using
the word significance, which is often associated with statistical parameters.

Agree to remove the use of the word significance to avoid confusion and to move the sentences as proposed.
The presence of debris cover glaciers is significant because gives an insight of the geomorphic evolution of the
cryosphere in the Dolomites The paragraph will be as following:

"Among the 51 glacial bodies, 13 are classified as mountain glaciers (Table 1) while 38 are considered snow or
ice patches (Smiraglia et al., 2015). When we use the term ice patch, we refer to the description of ice patch of
glacial origin present in Serrano et al. (2011), which is more specific and relevant to the study area compared to
the definition of dead ice. The presence of debris coverage is abundant or complete on 18 of the 51 inventoried
glacial bodies."

L106 Other Dolomites massifs that still host minor ice deposits devoid of any evidence of dynamics are not
included in this work.
What do you consider evidence of dynamics and how did you determine that none is present at these features
compared to the nine you study?

The previous sentence is based on Smiraglia et al. (2015) inventory work, a proper analysis of this is not
included in this work.

We will specify and add Smiraglia et al. (2015) citation in the text

Table 1: state in the caption or in the table for which year the area value is valid. Same year as the cited
publication?
Caption: Smiraglia and DIolaiuti → typo

The area for the Dolomites is valid for 2009, despite the work is from 2015. Also, the correct citation is Smiraglia
et al. (2015) and not Smiraglia and Diolaiuti, our mistake. We will add the year and correct this

**Data and methods**

Table 2: 2010 and 2012 photos have been used only for visual reference and not for mass balance reconstructions.
Would this be an opportunity to compare results using the 2010 photo vs. the 2010 lidar and assess the difference in elevation change between the different methods/data sources?

Even if this would be an interesting proposal, the problem is that 2010 surveys do not match in date and we have these kind of data only for one location (Mt. Antelao). Our proposal is therefore to improve the error and uncertainties section (see general comments) without including this comparison in the work.

L157: …using common area with regards to different years.
Unsure how to interpret this - does this mean you used the same area value for all computations of geodetic mass balance? Which area value (from which year) did you use?

We did not use the same area, but the common glacier area between each period. E.g. if the comparison is 2010-2014, we used their common area. Note that we have still reported both common and total area in the surface elevation change in Fig. 4 and Fig. 5

L157:
Every comparison included 2010 LiDAR data and has never been done using two historical SfM-point clouds at a time, to reduce possible sources of error. Does this mean you compared every other year to 2010? See general comment above.

Yes, we compared every other year to 2010. See comments above.

L163: ±0.12m I am assuming this refers to vertical accuracy? Consider clarifying In this study, our comparisons were done using relative distances; therefore, it may not be considered.
I don't understand what you mean here. Are you saying uncertainties in the lidar measurements are not considered? Please clarify why not.

It was unclear and as mentioned above this part will be implemented with a more robust accuracy estimation. All errors and their propagation will be now considered in the revised version of the manuscript

EM3C2 was available as a direct output of the algorithm (i.e., distance uncertainty), and considering our dataset was negligible compared to the $E_{AL}$.
So $E_{AL}$ was the main error source? Can you quantify the relative contribution of the different errors?

Yes, $E_{AL}$ was the main error source. As commented above we will provide a much more comprehensive evaluation of all errors and error propagation. We will also quantify the relative contribution of the errors

L170 imageries → imagery

Our mistake, will change it to imagery

**Weather station network**

L180 Additionally, years with missing data exceeding 5% of the accumulation (November to April) or ablation (June to August) season. Unusual definition of accumulation and ablation season, please explain the reasoning behind this. What happens in the missing months? (May, September, October)

Please, see comment "L199-L202"

L182 This was implemented at the level of individual AWSs, ensuring the availability of data for each year after averaging across all stations

Why average over all of them? If the goal is to get one T&P time series for the region, consider leading with
that.

The study area is small and individual stations do not show diverging trends among each other. While,
averaging among the stations allow us to have a more complete regional timeseries

L183 All the time series begin between 1985 and 2001 and end between 2020 and 2022 Does this mean that
none of the time series extend beyond 2022?

Yes they do, but when preparing the manuscript we stop to 2022 because more recent data were still not
available from the regional environmental agency. Furthermore, the study area is small and individual stations
do not show diverging trends among each other. Averaging among the stations allow us to have a more
complete regional timeseries

L189 where xa is either the total precipitation during the accumulation season (for the precipitation SAI, Pr SAI),
and the mean… Should this be "or the mean" ?

Yes, it is "or". Thank you

L191: The accumulation and ablation seasons were defined according to local climatology
Please specify what this means.

(Reply here refers also to comment to L180)

Nov-Apr is the time during which snow monitoring stations show increasing snow on the ground. From May the
mean snow on the ground start decreasing in all the snow monitoring stations. Dolomite glaciers are located at
lower altitudes compared to the Alpine average; therefore we used a shorter accumulation season.

L191: Finally, SAI values were spatially averaged, providing unique Pr and T SAI values for the entire region
Does "spatially averaged" mean you produced some kind of gridded data set or is this simply one averaged time
series over all the weather station data? Please clarify

It is the second one. "Finally, SAI values were averaged across all weather stations, resulting in unique Pr and T
SAI time series representing the entire region."

L193: The pre-processing applied to AWS data may result in an underestimation of total precipitation and
therefore of the Pr SAI.
Why? What part of the preprocessing leads to underestimation?

Due to the presence of missing data and the fact that years with more than 5% of missing data during the
accumulation or ablation season were excluded from the analysis.

L198 collect a datum
Consider rephrasing → record a value

OK, thank you. We will change it to "record a value"

L200: Using this data, we reconstructed the October to June snow depth on the ground for the most relevant
years of our study (1982, 1992, 2010, 2014, 2023).
Reconstructed as in you averaged over October to June for the given years? Or does the reconstruction involve
something more complex?

We have changed "reconstructed" with "show". The data we present here are daily snow on the ground as
recorded by the snow monitoring stations.

L202: Additionally, we calculated the October to June snow depth on the ground averaged over the whole time
frame for each station as well as the total annual snow accumulation.
Could you explain your reasoning for using October to June average snow depth? Wouldn't the snow depth at
the end of the accumulation season (late spring) be a more relevant metric?

We show the Oct to Jun (data are every 30 min or day according to the station) trend. In this way it is possible
to see the snow at the end of the accumulation season as well as the whole of annual trend. We rephrase the
sentence to make it clearer: "Additionally, we calculated the October to June snow depth on the ground
averaged over the five reference years for each station, as well as the total annual snow accumulation from
1980s to 2023."

**Results**

L206: Higher accuracy and precision (EAL 0.1-0.3 m) were obtained...
What does the EAL 0.1-0.3 m value represent? (accuracy or precision? Which years? What are the values in the
years where lower accuracy(?) was obtained?)

We will specify the highest and lowest accuracy and also the period considered in the revised version of the
manuscript. These 0.1-0.3m $E_{AL}$ values are referred to the recent comparisons (2010-2023) that do not include
analogue imagery. See also updates listed in the general comments answers.

L207 Out of the 9 glaciers analysed, Sorapiss Occidentale, Antelao, Marmolada and Pale di San Martino areas
were reconstructed starting from the 1980s while Popera and Cristallo reconstruction begins in the 1990s
State the exact years, 1980s and 1990s is vague

Agree, we will state the exact periods.

L214:
In 1980s and 1990s the Dolomites glaciers were larger in number, with several of them that have now
completely melted, turned into permanent ice patches without apparent ice dynamics and heavily buried by
debris.
Consider rephrasing for clarity. Something like: In the 1980s and 1990s, there were more glaciers in the
Dolomites, some of which have completely melted or turned into debris covered permanent ice patches
without apparent ice dynamics.

Agree, thanks for the feedback.

L217 Relative area reductions are not similar across all glaciers
State min max range of area reduction to show variation?

Agree, it's useful to present quantitative insights. Smallest area reduction is 9.1% in Popera Alto glacier while
largest is 88.9% in Fradusta glacier. Areas are also shown for all timesteps available in Table S2.

L219: topographic bounding
Consider explaining this term

Instead of using this term we will use "bounded by steep topography" to be clearer

L226: for common and total glacier area
Please explain what you mean by common and total area. Is this stated somewhere?

See general comment "Glacier Area". We will add a specific explanation that was now missing.

L226: Due to the impossibility of retrieving enough data for years 1999 and 2001, we considered the period
from 1990s to 2010 as a unique time frame, instead of calculating the metrics at a decadal frequency. The average cumulative surface elevation change (Table 3) was calculated for three periods: 1980s with -5.21 m,
1990s-2010s with -14.09 m and 2010s with -9.31 m.
Does "unique time frame" just mean you used a longer time step? I think rephrasing would help clarify this,
something like: "The average cumulative surface elevation change (Table 3) was calculated for three periods:
1980s with -5.21 m, 1990s-2010s with -14.09 m and 2010s with -9.31 m. Due to lack of data in 1999 and 2001 it
was not possible to resolve the 1990s-2010 period at decadal frequency."

We agree with the proposed rephrasing that avoids potential misunderstanding.

L241 The highest absolute losses, corresponding to almost 35 m, are reached in the area involved in the ice
avalanche that happened in a detached part of Marmolada Principale, on 3$^{rd}$ July 2022, as shown by the Kernel
Density plots of surface elevation loss (Fig. 5b)
Can you mark this in the figure? I am unsure where I can see this in Fig 5b.

As this is under "Marmolada Collapse" label in Fig. 5b, we will add a reference to it in the text to help the
readers finding it in the figure. The same label is present also in the map (Fig. 5a) so it should be easy to find it.

L243 The Fradusta Inferiore Glacier was not included in the common area measurements as it had already
disappeared before 2023 surveys took place.
Again, what exactly is common area?

See comments above on Glacier Area. More explanations will be added in the methods.

L246 On that glacier a rise of more than 10 m has been observed close to a wide serac whose presence is
possibly related to a small surge induced by a recent rockfall (Fig. 5a) in the accumulation area as well as by
internal glacier dynamics
Interesting! If possible, consider marking this feature in the figure

This feature is already shown in Fig. 8a, b and more text is present in the Discussion section.

L248 This is well visible in Fig. 6a,
Should this be Fig 5a?

You are correct, our mistake

L251 ff and Table 4:
Do these uncertainties refer only to the uncertainty originating from the density conversion, or does this also
include uncertainties in area and volume?

Uncertainties in area and volume were not present and will be updated in the revised version of the
manuscript. See general comment on "Surface change computation and treatment of errors".

L256 Our results show that the use of a fixed maximum glacier area in the geodetic mass balance leads to an
underestimation of the m. w.e. loss when compared to common area calculations. In our case the bias
introduced by total area is between -1% and -31% of the common area mass balance, depending on the site
and considered period. There are some cases of decadal comparison (1980s-2010 in Cristallo, Antelao Inferiore
and Marmolada) where total glacier area produced larger mass balance losses than calculations using common
area.
I am still unsure about the differences between "fixed maximum glacier area" (this term is used for the first
time here), common area, and total area.

See comments above on Glacier Area. More explanations will be added in the methods.

Table 4: (a) Sorapiss Occidentale values have been corrected removing the positive elevation gain portion for
2010-2023.

Why did this need to be corrected? Did you simply delete all positive values or was there some other
correction? You measured the positive elevation change and suggested that this was due to a rockfall/surge
process - what is the argument for removing the elevation gain when that is what your analysis shows?

We have removed the positive elevation change values (simply removing values > 0) from Sorapiss glacier to try
to get a more realistic estimation of the mass balance rate. Although this is not the most precise evaluation, we
think is still better than showing the mass balance rates including that positive values.

We agree that the positive change (in surface elevation change) measured should be shown, and that is why in
Table 3 we did not apply any correction.

**Climate data**

L266. Among the ten highest events, seven have occurred in the last 15 years (2007-2022).
Consider rephrasing for clarity? highest → warmest

We would prefer to use "high-low" as we are actually writing about SAI and not T, even if high SAI means
warmer T.

L269 The maximum Pr SAI has been calculated for 2014 with a value > 2, while 1996 is marked by the minimum
value at -1.22.
Consider rephrasing for clarity, e.g.: Pr SAI was greatest in 2014 with 2.x and lowest in 1996 with -1.22.

Ok, thank you. We'll rephrase. "Pr SAI was greatest in 2014 with a value > 2 and lowest in 1996 with -1.22."

L272 Temperatures have risen by 0.4-0.6 °C per decade since 1985, while precipitation showed an increase that
lasted about 15 years from 1995, culminating in the extremely snowy year of 2014 (Fig. 6b). Fedaia station, the
only one providing data since 1980, does not show any trend for the total snow accumulation (p-value = 0.61;
Fig. 6c), however, increased extreme events can be observed in the last decade of its time frame. The other
three snow monitoring stations exhibit slightly different patterns, demonstrating a higher frequency of snowy
winters also in previous decades. Did you also look at station variability for T and P? How do you identify
extreme events in the snow time series?

P and T trends among stations were similar. Furthermore, since the study area is quite small and none of the
stations is on/adjacent a/to a glacier we preferred to use regional mean values. Extreme events are considered
those events falling above the 95th percentile.

We will modify the text consequently: "The snow monitoring stations, do not show any trend for the total snow
accumulation (p-values = 0.54-0.95; Fig. 6c), however, extreme events (above 95th percentile) were observed in
2013 and 2014 for all the stations. "

Fig 6b: The dotted line is hard to see. I'm assuming the lines refer to hydrological year, i.e. 2023 refers to the
2022/23 winter season. Consider stating this in the caption or legend.

We will modify the caption: "…for the same snow monitoring stations (c). The years shown in the plot refer to
hydrological years, e.g. 2023 refers to 2022-23".

**Discussion**

L295 In the Dolomites, a slight increase in winter snowfall has been observed at some high-altitude stations,
such as Ra Vales site at 2620 m (Fig. 6)
How do you determine this increase? It is not really obvious from Fig 6c and there is no mention of this in the
results.

This slight increase has been determined since 1993 using linear regression and is present in all the 4 stations, but more evident for the highest one (2620 m a.s.l.). Extending the linear regression from the beginning of the time series (i.e., 1980 and 1987) bring slightly different results, even with slightly negative values. We will add this to the result as it was absent and implement this part also in the discussion.

L296 unfavourable years conditions for glaciation prevailed
Extra word? Delete "years"

Yes, it was a mistake

L302 Within Alpine mass balance records, the ablation season of 2022 results unprecedented.
Missing word? (…results were unprecedented…)

Yes, we will correct it adding "were"

L305 According to such climatic evolution, the Dolomites are rapidly turning from being mountains hosting sites favourable to local glaciation, to areas where peri-glacial processes will progressively gaining importance.
→ gain importance

Agree, thank you for the correction

L317 Dolomites glaciers mass balance rates are half of the average RGs rate during the last 13 years
Interesting!

L334 stabilise the dynamic of some glaciers of the Dolomites
Do you actually mean dynamic as in movement or something else? Consider rephrasing

We meant that extremely snow winters like 2014 can stabilize the mass balance of the Dolomites, as glaciers, as shown in the 2010-2014 comparisons. The sentence needs to be rephrased to avoid confusion.

"The occurrence of extremely snowy winters can still result in an increase of volume for some glaciers in the Dolomites. This is evidenced by our data from at the end of summer of 2014, when 5 glaciers of the Dolomites (Popera, Sorapiss, Antelao Inferiore, Marmolada) have recorded a positive cumulative mass balance since 2010."

Fig 7: Cool figure! I'd be interested in seeing how the WGMS annual product compares to your values for the Dolomiti glaciers (just an idea, the figure is informative as is and this is not needed for the manuscript)

https://cds.climate.copernicus.eu/cdsapp#!/dataset/derived-gridded-glacier-mass-change?tab=overview

We have tried to compare the values we had with WGMS annual product (see Fig. R2). 2023 is missing from the available WGMS annual gridded products and we don't think that considering the high local variability of our very small glaciers it is worth adding it to the figure or the paper.

Fradusta (-12.8 m w.e.), Travignolo (-8.5 m w.e.) and Marmolada (ranges between -14.5 and -8.4 m w.e.) are in a WGMS cell of -6.76 m w.e.

Antelao Superiore (-10.0 m w.e.) and Inferiore (-6.4 m w.e.) are in a WGMS cell of -4.27 m w.e.

Sorapiss (-3.8 m w.e., with correction, see comments above), Cristallo (-8.6 m w.e.), Popera Alto (-7.4 m w.e.) and Pensile (-5.8 m w.e.) are in a WGMS cell of -10.85 m w.e.

[Figure]

Figure R2. WGMS Annual Product Grid sum from 2010 to 2022 and position of the analyzed glaciers.

L360 most representative
If it is the largest it is not the most representative in terms of size. Consider removing this.

Correct, we will remove the "… and most representative …" as it's not correct. We meant it's the most representative for Alpine scale or WGMS comparisons and of course in eventual weighted means.

L374 In this study we used the surface lowering observed during the last 13 years and direct observations on site to assess the glaciers end.
I would like to read this earlier, e.g. in the methods.

Agree, this sentence will be put in the methods at L141 and slightly modified to fit in the paragraph. We will add also a sentence to specify how glaciers area have been mapped as also requested by R2. As per now is only mentioned in L141.

L397 In the late 1950's the Dolomites were hosting 33 glaciers, of which only 9 are still active;
Define somewhere what you mean by active

As mentioned in the comments above here we refer to Smiraglia et al. (2015) Italian inventory. We will specify this "… of which only 9 are still considered mountain glaciers (Smiraglia et al., 2015).

L402 A few glacial bodies may eventually shift from glacial to periglacial, thus becoming more resilient in a warming climate.
There seems to be an ongoing discussion about how and whether glacial features can turn into periglacial features (e.g. discussion comments here: https://tc.copernicus.org/articles/18/1669/2024/tc-18-1669-2024-discussion.html ). Perhaps rephrase this sentence to avoid ambiguity. You could focus on the processes that would make the ice features more resilient without classifying them as glacial or periglacial.

We partly agree on this, despite in the Dolomites region these glacial-periglacial shift appears as an ongoing phenomenon (see e.g Seppi et al., 2014). It is anyway a good idea to rephrase the sentence to avoid ambiguity

---

## Author Comment (AC2)

We have replied to the comments in blue.

**Review of 'The Glaciers of the Dolomites: last 40 years of melting'** By Securo, Andrea and others.

Securo and authors present a multi-decadal estimation of surface elevation change for small glaciers in the
Dolomites, Italian Alps. Their geodetic data used in this study consists of aerial photographs, uncrewed aerial
vehicles (uavs) and LiDAR data. This data and their analysis indicate high rates of glacier mass loss with the
Marmolada Glacier accounting for about ⅔ of the region's volume loss.

Overall, I found this manuscript to be generally well written with methods partially described. The presented data
generally supports the conclusions made by the authors. However, like many papers, some clarification of the
methods is needed, the English can be substantially improved, and manuscript could be shortened. Below, I
outline my major points about the paper and follow these with technical comments.

MAJOR POINTS:

1. Methodology and error analysis needs further description - Given that the authors are using multiple geodetic
datasets to calculate volume (and mass) change, the methods section should clearly lay out how the actual
uncertainties are propagated to yield the total error budget and final uncertainties. If I understand it correctly,
the authors obtained point clouds from all datasets, align these and then use M3C2 for the error analysis. There
has been some discussion about the robustness of M3C2 as it assumes planar surfaces and could estimate
significance of detectable change (e.g. https://doi.org/10.1016/j.isprsjprs.2021.06.011). I would like to see, for
example, how the uncertainty over stable terrain changes due to slope. These are small glaciers in rugged terrain
so their slopes will be steep. Also, are there assumptions made for missing data? How is this error source treated?

As reported in the technical comments and suggested by both Referees, we agree that the error analysis needs
to be better clarified and improved in a few points, especially regarding error propagation which can be done
better. We do not think section 3.3 needs further addition, it is more about the clarity of the text and the approach
used, that we will improve in the revised manuscript. We will consider all errors from LiDAR, point clouds
alignment and M3C2 distance measurements in the surface elevation change measurements. See also comments
from Referee #1.

These small glaciers are in rugged terrain and have high slope values, we agree with the Referee that M3C2
uncertainty should be referred to areas with similar slope of the glaciers. And that's why we have done that. We
will write a few more details about it on the methodology. We used verticality (geometrical attribute in
CloudCompare) to measure the slope of the glaciers and selected only areas with similar values on stable terrain
to calculate the alignment error ($E_{AL}$). We will implement the revised manuscript with more text and more
information on how the uncertainty over stable terrain changes due to slope, as requested. This information
specifically will be put in the supplementary materials.

In the case of completely missing data, we have not performed any kind of interpolation. This is mainly because
the portions of no data are very small compared to the reconstructed areas and are assumed to be in line with
the resulting average. We will specify better this in the reviewed manuscript. Just to be clear, missing data were
present only in very small portions of Marmolada 1982-2010 and Antelao 1982-2010 (Fig. 4) and Popera Alto
Glacier 2010-2023 (Fig. 5a). An additional part where noise in data was present is Marmolada Principale 2010-
2023 (Fig. 5a). The influence of these voids in the final average result for surface elevation change is not relevant.

2. Manuscript needs to be shortened/tightened - I found the 'Introduction and Previous glaciological research'
sections to be long and would strongly advocate for merging the 'Previous glaciological section' with the Intro so
that the total length of both sections is about ½ of what it currently is. Also, I think some of the tables could be
moved to a supplement as most people rarely need to read each line of these tables (they are, however, useful
to have if a reader needs them).

While we agree with reviewer 2 that the manuscript appears long in some of its sections, we do not agree with
the proposal of shortening sections 1-2 by approximately 50%. In particular, the second chapter that deals with
the previous glaciological research in the Dolomites was made because there are no previous recent work dealing
with the evolution of these glaciers. We think it can be insightful and potentially useful, especially out of the
Italian/Alpine community, to read and find a recap of the previously available glaciological research in the area,
which often is not easy to find and is almost entirely written in Italian and only available through grey-literature,
local chronicles, and regional reports. Therefore, we propose only some smaller shortenings to the introduction,
without changing chapter 2. The only additional paragraph that repeats information between introduction and
chapter 2 is from L40 (… In the 1960s, the surface of…) to L46, and will be removed.

One possible alternative option, on which we ask the editor's opinion since Referee #1 did not object to the length
of the manuscript, is indeed to move the section on previous glaciological research in the Dolomites (Chapter 2)
to Supplementary Materials.

3. Comparison to previous work - The authors do a commendable job compiling datasets for these small glaciers,
but they should explicitly show how their results compare to those, for example, of Hugonent and others (2021).
The authors can download data for each of their glaciers (since each glacier has an RGI number this should not
be a difficult task). How do their estimates and uncertainties compare to Hugonnet? This is an important test of
the reliability of Hugonnet for small glaciers in this region (I would posit that perhaps the present study has better
estimates for these small glaciers but I simply don't know). It would be good to examine this in some detail.

Hugonnet et al. (2021) work is certainly valuable and precise for larger glaciers, but in this specific study area the
resolution is simply not enough to evaluate the surface elevation changes correctly as we did using a higher
resolution dataset. As you can see from the figure below (Fig. R1) there is a different order of resolution (pixel
size of Hugonnet et al. data is 100 m) between our calculations the one in Hugonnet et al. (2021), due to their
much larger effort in terms of total area. Comparing our results with those would therefore not be useful and can
be misleading, as shown in the example below.

We will add a sentence in the discussion about this issue as it can be valuable information. Possibly with future
studies based on much higher satellite remote sensing imagery (e.g. Airbus Pleiades) this gap can be reduced,
and the values can be compared. Furthermore, there is a problem with the difference in the timesteps of the
comparison that do not match between our work and Hugonnet et al. (2021).

[Figure]

**Fig. R1.** Example of Hugonnet et al. (2021) surface elevation change dataset from 2010 to 2019 (m) in the Dolomites Glaciers.

4. Manuscript should be shortened. I found the length of the discussion section to be somewhat unbalanced with
the length of the results section (the former is longer than the latter). I would recommend shortening the
discussion section to balance in light of statements that can be backed up with results shown in the results
section. Some of the figures/materials in the discussion seem, to me, to be more results and less discussion.

It should be noticed that the discussion are this long because of the first part regarding climate in the area

We agree regarding the fact that part of Discussion material can be moved to Results and will do that.

Furthermore, section 5.1 dealing with climate in the Dolomites region could be of less relevance considering the
aims of the paper compared to 5.2, where the results are compared to WGMS reference glaciers. We will try to
shorten all the three sections leaving more relevant parts only.

We can move Fig. 8 and eventually Fig. 9 to Results.

5. Terminology needs clarification - I would recommend that the authors consult the glossary for the standard
definitions used in glacier mass balance (https://wgms.ch/downloads/Cogley_etal_2011.pdf) . There are multiple
appearances of mass balance when the authors are referring to 'geodetic balance'.

We agree with reviewer 2 on this point, especially regarding the term "geodetic balance".

TECHNICAL COMMENTS:

Abstract:

'Use lowercase 'Alpine' unless it starts a sentence.

Pg 1, Line 3: This sentence isn't technically correct in light of Worldview or Pleiades (very high resolution)

Pg 1, line 11: 'between used for two items, among for more than two'. Also use of 'amplitude' is vague

Pg1, line 13: replace 'areal reductions' with 'area loss'

Pg1, lines 13-15 - This sentence is out of place and likely not needed

P1, line 18: 'with greatest emphasis on regions of the world' - unclear what authors are referring to here.

We agree with Reviewer on the proposed corrections for Pg 1 and changes of the sentences of introduction

We will remove the sentence of L 13-15

Regarding L18 we mean that regions like e.g. the European Alps, and the Arctic global warming hotspots and here
increased ice losses rates are reported. We can reformulate for more clarity as: "Glaciers worldwide have been
losing mass at alarming rates over the past decades (Zemp et al., 2019; Hugonnet et al., 2021). This is particularly
evident in regions where warming is occurring at a faster rate than the global average (Rantanen et al., 2022;
ICCI, 2022)."

Pg 2, lines 25-33 - Authors could easily jettison this section to shorten first two sections that need to be put on a
diet.

Agree on reduction despite this being the first paper dealing with this study area makes it hard to reach the
proposed shortage of 50%. We think, as written in the general comments, that an overview of the previous
glaciological research in the Dolomites is useful. Regarding these specific L25-33 we think they introduce the
Dolomites region (to eventual people that don't know it) and specify why it's an important area. We will shorten
the paragraph a bit, reducing it to 2 sentences.

Pg 2, line 50 (and throughout) - 'unmanned' is an outdated term these days. Typical use is 'uncrewed aerial
vehicle'

Will change the terminology throughout the text pg.3 , line 63. Unclear what authors mean with 'active' - deformation, ice flow?

For active we mean that these have been classified as mountain glaciers in the latest available Italian glacier
inventory (Smiraglia et al., 2015). We will specify it better in the text

Section 2 - Shorten this section

See general comments above on Section 2

Figure 1. The figure could use a little bit of work. The inset (upper left) needs at least some lat/lon coordinates
for a reader not certain where the Dolomites are. I was initially confused with the color of the glaciers and the
colors of the geodetic datasets (numbers). Maybe change the color of glaciers to avoid confusion? (a) - replace
'position' with 'location'

We will improve the figure readability, and we will add coordinates and adjust the colors/labels as proposed

Pg. 6, lines 115-118. Were these photos not available as photogrammetric scans? Also, it's too bad that the
internal orientation (if available) information isn't used as that might help reduce overall error budget.

Unfortunately, the photos used were not available as photogrammetric scans and internal orientation was not
available either. These problems are also evidenced in the discussion. We will add in the sentence that also
internal orientation was not available

Pg, 6, lines 115-126. I had a hard time understanding how GCPs were collected and how they were used. This
section should be revised to make it clear exactly what was completed and for which datasets.

We agree this should be clarified better in the text and we will improve it. Perhaps Fig. R2 (see below) that will
be put in the supplementary materials could be helpful for understanding the methodological steps more easily.

Ground Control Points have all been taken from 2010 LiDAR dataset, which I the best available so far, and used
for all SfM processing. The entire procedure of retrieval of GCP was done in CloudCompare, while GCPs
coordinates have been used in Metashape during SfM processing.

[Figure]

**Figure R2**. Summary of the data processing proposed to be integrated in the Supplementary Materials.

Pg. 8, lines 153-154. I'm surprised that there were only small areas of voids. Was hypsometric interpolation attempted?

The areas of void were very small without hypsometric interpolation, as stated in the method we used Metashape meshes instead of dense point clouds when the voids were too big to fill. Metashape in this processing does a default linear interpolation and works only within a certain radius depending on the resolution of the point clouds. From our visual inspection this does not introduce any change in the dataset and works as a simple linear interpolation

We think that the areas of voids are this small because photos have a lot of details and are not shot from very high altitudes, despite not having the photogrammetric scans. The areas of the glaciers with snow featureless pixels or are not that impacting in the study area, at least not in these dataset

Pg 11, lines 210. Area change. How were areas of the glaciers digitized? Any uncertainty in area change? Unless I missed it, planimetric mapping is not described in methods.

Areas were manually digitized but we can better explain the procedure especially in regard to the role of debris in the area evaluation. For this reason, we propose to improve the current paper adding the presence of debris in 2023 areas to current Fig. 3 to let the reader know where the glacier areas could be susceptible to major uncertainties due to the debris cover

Pg. 11, lines 219: Unclear what 'topographic bounding' is. Surrounded by rugged terrain?

Yes, that's what we meant; we will specify better in the text

Pg. 11, lines 228-229: Uncertainties needed for these estimates.

Agree, see previous comments on the uncertainties improvement

Pg. 13 lines 215: Terminology needs to be changed to include term 'geodetic'

Agree, see answer to general comments 2 and 4, above

Figures 3, 4 - Generally well drafted, but uncertainties would be useful.

Uncertainty for area are harder to quantify. As stated above, we propose to implement the area figure with debris
cover but without quantifying the uncertainty of this, lacking geophysical data. This problem is already specified
in the discussion.

Uncertainty for surface elevation change (Fig. 4-5) are shown for each comparison in the bottom right corner
(grey).

Tables 3, 4- I would recommend moving these perhaps to a supplement. Also it would be good to have a summary
line for weighted mean (table 3). Does table 4's all glaciers line imply this is a weighted mean (by area)?

We agree with the Referee, and we suggest to move Table 3 to the Supplementary Materials. Table 4 in our
opinion should stay in the manuscript.

Figure 5 - I think this is a well drafted figure, but I'm not fond of the color bar. It really should be a standard
diverging color (red to blue). The dark ends for both red, blue are problematic for diverging data. Sorapiss Glacier's
mid elevation I presume is debris covered? I would explain before it is brought up in the discussion.  As stated in
the major comments, I think some of the discussion and plots should be moved to results section.

The color bar is made with darker ends to improve the spatial visualization of largest changes, which would
otherwise appear flat and not be distinguishable in the maps (see Fig. R3). The same color scale is used also in
the previous Fig. 4 for the same reason. We have done some tests and can report here an example for Marmolada
to show what we mean. Dark ends can be problematic, but they are still distinguishable especially in this case
where the positive "blue" extreme (+ 20m) is not present on the map.

Regarding the debris cover, we think improving Fig. 3 with the current presence of debris cover in 2023 for all
glaciers would help the overview of area change and current situation of the glaciers. Will produce a revised
version of the Figure including debris cover. This should clarify also Sorapiss current debris coverage (visible also
in Fig. 8b).

Regarding the Figures in the Discussion-Results sections, we think Fig. 8 can be moved to Results while Fig. 9 is
more a subjective choice.

[Figure]

**Fig. R3.** Example of the chosen diverging color scale with darker colors at the extremes and the standard diverging color scale.

Figure 7. Not certain what color bars on top of the graph refers to. WGMS trend is the dashed line? It's not
evidently clear to me.

The colors on the top graph pf the bar refer to Reference Glaciers (gray) and to the Dolomites glaciers, following
the color scale they have on the plot. We think it's clear but to avoid confusion will specify this in the figure
caption. WGMS average is the only text in black and the only line in black, looks clear to us but will use a thicker
black line without dashes to improve visualization.

Pg 16, Results - How do these results compare to Hugonnet? Add those values perhaps to one of your tables with
both estimates (yours and Hugonnet) including uncertainties.

See the general comment above and related figure example on why we think this comparison is not that useful
considering the small size of the glaciers of the Dolomites.

---

## Author Response (AR1)

Dear Editor and Reviewers,

Please find the tracked changes and new version of the manuscript and supplementary materials.

The major and detailed comments of both reviewers were answered in the previous AC-1 and AC-2 files and corrected during the revision.

- Answer to comments from Referee #1:
  https://egusphere.copernicus.org/preprints/2024/egusphere-2024-1357/egusphere-2024-1357-AC1-supplement.pdf
- Answer to comments from Referee #2:
  https://egusphere.copernicus.org/preprints/2024/egusphere-2024-1357/egusphere-2024-1357-AC2-supplement.pdf

In the updated version of the manuscript, we have done the following major changes:

- Revised the description and methodology for uncertainties, including all of them into a more robust evaluation
- Reduced the length of the manuscript, moving the tables to the supplementary materials as most of their information were already present in figures or in the text. We have also worked in shortening Section 2 and 5, as proposed by #R2
- Revised Fig. 1, 3, 4, 5, 6 following #R1 and #R2 comments and feedback
- Assessed all minor comments and corrections throughout the manuscript, see AC-1 and AC-2 above
- Implemented the description of the methodology and added a scheme in the supplementary material (Fig. S1)

Best Regards,